# Effect of Lithological Parameters on Combustion Characteristics of Methane Hydrate Sediments

**Gan Cui [1,\*], Di Wu [1], Yixuan Li [1], Shupeng Yao [2], Tao Guo [3], Juerui Yin [1], Xiao Xing [1] and Jianguo Liu [1,\*]**

[1] College of Pipeline and Civil Engineering, China University of Petroleum (East China), Qingdao 266580, China; wudiliyixuan@gmail.com (D.W.); liyixuanwudi@gmail.com (Y.L.); jueruiyin0721@foxmail.com (J.Y.); 20170079@upc.edu.cn (X.X.)

[2] Qingdao Qinggang Tongda Energy Co., Ltd., Qingdao 266580, China; fwfzajbysp@sd-port.com

[3] Nanjing Metrology Research Center, PipeChina West East Gas Pipeline Company, Nanjing 210046, China; guotao02@pipechina.com.cn

[\*] Correspondence: 20170042@upc.edu.cn (G.C.); liujianguo@upc.edu.cn (J.L.)

**Abstract:** In this paper, four lithological parameters, that is, thermal conductivity, particle size, porosity, and saturation, are investigated by combining experimental observations with numerical simulations to study the influence laws and mechanisms of action on the combustion process of methane hydrate sediments. The variations in combustion characteristics parameters such as flame height, effective combustion ratio, and dimensionless discharge water mass with the lithological parameters are studied. In addition, the combustion mechanism of lithologic parameters on methane hydrate deposits is revealed. Combining the experimental results and simulation calculations to optimize the combustion of methane hydrate sediments, it is recommended to use methane hydrate sediment samples with high saturation and low thermal conductivity, while the oxidant concentration and porosity of methane hydrate sediment samples should be increased.

**Keywords:** lithological parameters; methane hydrate sediments; combustion characteristics; numerical simulation

## 1. Introduction

Natural gas hydrate is a cage-like compound formed by natural gas and water molecules under high pressure and low temperature [1,2]. Natural gas hydrate is considered one of the most important alternative energy sources in the 21st century because of its huge reserves, high energy density, low carbon emissions, lack of harmful substances, etc. [3,4]. The exploitation of natural gas hydrates is of great significance to solving the problem of future energy supply [5,6].

The principle of natural gas hydrate extraction involves disrupting the thermodynamic equilibrium of hydrates through various methods, thereby causing the decomposition of hydrates and the release of natural gas [7]. To date, numerous countries have conducted experimental hydrate extraction [8]. However, the challenges posed by the high-pressure, low-temperature, and low-permeability subsurface conditions as well as the complex and variable decomposition characteristics make hydrate extraction a difficult endeavor. Achieving economically viable, safe, and efficient extraction of natural gas hydrates remains a focal and challenging issue both domestically and internationally [8,9]. Presently, the conventional extraction methods that have been proposed include depressurization, heat injection, displacement, and inhibitor injection methods. However, these methods come with drawbacks such as high energy consumption, low productivity, geological disruption, and environmental pollution, which constrain the commercial development of hydrate resources [10]. To further enhance extraction efficiency, researchers have introduced new extraction methods, including dual-horizontal well thermal water injection, single-well thermal drawdown, and electrically assisted depressurization methods [11].

In recent years, the in-situ combustion method has gained the attention of researchers due to its advantages, such as low heat loss, and it is proposed that it can be applied to the exploitation of natural gas hydrate [12–16]. Furthermore, directly utilizing the natural gas hydrates within the reservoir as fuel can help mitigate the limitations of in-situ combustion and further advance its commercial application [17]. However, existing processes still require the external transportation of fuel and oxidizers into the combustion heat exchanger, involving on-site fuel storage and injection, which increases technical complexity and costs. By conducting combustion experiments on methane hydrates, scholars [18–21] have found that the energy transfer between the combustion flame and the hydrate decomposition products is uneven and unstable. Direct combustion of natural gas hydrates leads to lower flame temperatures and lower combustion efficiency [22,23]. In addition, the combustion of hydrate sediments has the disadvantages of poor sustainability, instability, and low heating value. From the current studies, the main parameters affecting methane hydrate combustion include effective area for heat supply, external gas convection, characteristic thickness of the powder layer, heat flux, moisture concentration in the combustion, and powder reservoir geometry [24,25]. Therefore, it is imperative to further study the combustion characteristics of natural gas hydrates in order to elucidate the optimization mechanisms of their combustion and provide corresponding optimization recommendations.

In the previous work, the problem of optimizing external conditions for the combustion of methane hydrate sediments was studied [17]. Maruyama [26] et al. investigated the effect of temperature on the flame propagation rate of methane hydrate. Nevertheless, the optimization of internal conditions for the combustion of methane hydrate sediments has not been investigated. Lithological parameters are important physical parameters of methane hydrate sediments, which play a decisive role in many processes [10,27]. Therefore, studying the mechanism of the effect of lithological parameters on the combustion process of methane hydrate sediments will be useful to solve the problem of optimizing internal conditions for the combustion of methane hydrate sediments.

The novelties of our current work include the following scientific and application aspects. This paper investigates the combustion characteristics of methane hydrate sediment under different lithological parameters and analyzes the impact patterns and mechanisms of these parameters on the combustion process. The image of flame evolution and the data on the combustion characteristics of methane hydrate sediments were obtained through combustion experiments. The influence of four lithologic parameters, such as thermal conductivity, particle size, porosity, and saturation, on the combustion evolution and combustion characteristics of methane hydrate sediments was analyzed.

Through numerical simulations, the temperature field distribution at transient moments during the combustion of methane hydrate sediment and the concentration field distribution of key components were obtained, and the influence mechanism of different lithology parameters on the combustion process of methane hydrate sediments was clarified. Finally, according to the research conclusions, the combustion optimization suggestions for methane hydrate sediments with different lithology parameters are put forward. The results of this study contribute to enhancing the combustion efficiency of methane hydrate sediment and offer new insights for the efficient application of in-situ combustion recovery methods.

## 2. Experimental Research

### 2.1. Experimental Parameters Determination

The lithological parameters selected for the experiments in this paper and their corresponding value ranges are shown in Table 1. It should be noted that the main physical difference between the different gravel types is the thermal conductivity, which is white corundum > brown corundum > quartz sand. The methane hydrate particle diameters prepared in the laboratory are about 0.45–0.5 mm. Therefore, the relative ratios of gravel particle diameters to methane hydrate particle diameters range from 1 to 4. Porosity is

defined as the ratio of pore volume to total volume. Saturation is defined in the same way as pore saturation, that is, the ratio of methane hydrate volume to pore volume.

**Table 1.** Summary table of the experimental parameters selected in this paper.

| Types of Lithological Parameters | Range of Lithological Parameters |
|---|---|
| Particle size | 0.45–0.6 mm, 0.9–1.18 mm, 1.43–1.7 mm, 1.7–2.0 mm |
| Porosity | 0.3, 0.4, 0.5, 0.6 |
| Saturation | 0.4, 0.5, 0.6, 0.7 |
| Gravel type | Quartz sand, brown corundum, and white corundum |

*2.2. Sample Preparation*

In the experiment, methane hydrate sediment samples are prepared by the method of in-situ generation and mix sample preparation, that is, the methane hydrate samples are first prepared and then mixed with gravel to form methane hydrate sediment samples. The formula for calculating the mass ratio of methane hydrate samples to gravel samples is derived from reference [17] and is shown below:

$$\varphi = 1 - \frac{m_s}{V\rho_s} \tag{1}$$

$$S_H = \frac{m_H}{\varphi V \rho_H} \tag{2}$$

$$\frac{m_H}{m_y} = \frac{r_g}{r} \tag{3}$$

$$\frac{m_s}{m_y} = \frac{(1-\varphi)r_g\rho_s}{r\varphi S_H\rho_H} \tag{4}$$

where $\varphi$ is the porosity; $m_s$ is the mass of mixed gravel, g; $V$ is the total volume of sample, m³; $\rho_s$ is the density of mixed gravel; $S_H$ is the methane hydrate saturation; $m_H$ is the mass of pure methane hydrate contained in the methane hydrate sample, g; $\rho_H$ is the methane hydrate density, 0.9 g/cm³; $m_y$ is the total mass of the methane hydrate sample, g; $r_g$ is the mass fraction of methane in the methane hydrate sample; $r$ is the maximum gas content of methane, which can be calculated from the chemical formula of methane hydrate $CH_4 \cdot 5.75H_2O$, 0.1339.

To ensure the credibility and generality of the experimental laws, the preparation of methane hydrate samples must be reproducible. In this experiment, the gas content of methane hydrate samples is calculated by the weight difference method to characterize their reproducibility. Table 2 shows the gas content of methane hydrate samples from different prepared batches, and it can be seen from the table that the methane hydrate samples prepared in this experiment have good reproducibility.

**Table 2.** Gas fraction measurement of methane hydrate samples.

| Batch | Number | Pre-Reaction Mass/g | Post-Reaction Mass/g | Methane Mass/g | Gas Content | Average Value of Gas Content of the Batch |
|---|---|---|---|---|---|---|
| | 1 | 18.43 | 19.52 | 1.09 | 0.0978 | |
| 1 | 2 | 17.29 | 18.25 | 0.96 | 0.0970 | 0.0973 |
| | 3 | 17.12 | 18.05 | 0.93 | 0.0963 | |
| | 4 | 16.3 | 17.15 | 0.85 | 0.0965 | |
| 2 | 5 | 17.28 | 18.23 | 0.95 | 0.0963 | 0.0972 |
| | 6 | 16.6 | 17.45 | 0.85 | 0.0951 | |
| | 7 | 19.88 | 21.1 | 1.22 | 0.0958 | |
| 3 | 8 | 17.85 | 18.85 | 1.00 | 0.0953 | 0.0971 |
| | 9 | 19.28 | 20.41 | 1.13 | 0.0939 | |

### 2.3. Experimental Setup

Figure 1 shows the schematic diagram of the combustion experimental setup built in this paper based on references [17,21,28–30]. The combustion experimental setup mainly consists of a combustion test unit and a data acquisition unit. The flaring test unit consists of a methane hydrate sediment sample (3), a combustion tank (4), a stand (5), and a thermocouple (7). The data acquisition unit consists of a camera (1), a load cell (6), and a data acquisition module (8).

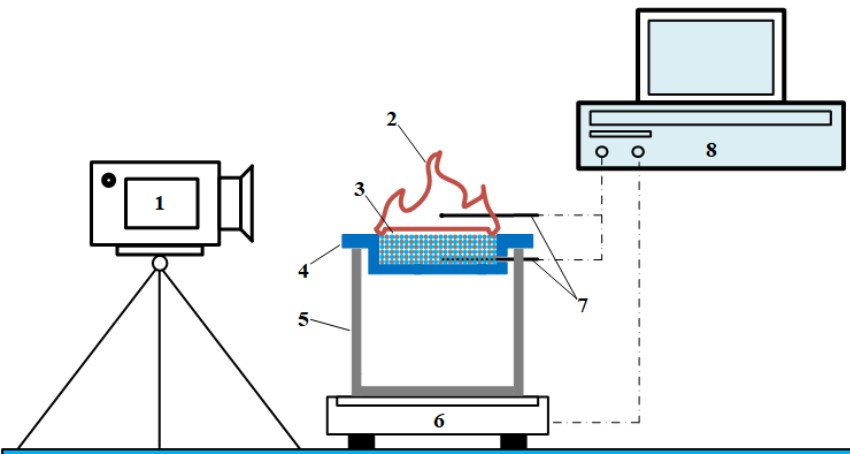

**Figure 1.** Schematic diagram of the combustion experiment setup. 1—camera, 2—sample burning flame, 3—methane hydrate sediment sample, 4—combustion tank, 5—stand, 6—load cell, 7—thermocouple, and 8—data acquisition module.

### 2.4. Definition of Combustion Parameters

To evaluate the combustion efficiency of methane hydrate sediments under different lithological parameters, several combustion characteristic parameters are defined based on references [17,21,28–30]. The effective combustion ratio is defined as the ratio of the amount of methane burned in the sample to the total amount of methane contained in the sample, which is shown in Equation (5). In addition, to characterize the rate of mass change in the sample during the combustion process, the change in the combustion ratio per unit time of the sample is defined as the ratio of the effective combustion ratio to the burning time, and its mathematical expression is shown in Equation (6).

$$\eta = \frac{m_0 - m_z}{r m_0} \tag{5}$$

$$m^* = \frac{m_0 - m_z}{r m_0 t} = \frac{\eta}{t} \tag{6}$$

where $m_0$ is the initial mass of the filled sample before combustion, $m_z$ is the final mass of the filled sample at the end of combustion, and $r$ is the methane mass fraction of the sample.

Figure 2 shows the binary image of the flame with the help of MATLAB. In the binarized image, the transient flame height (red dashed line) is defined as the difference between the highest point of the flame and the upper wall of the combustion tank (blue dashed line). To facilitate the comparison of different combustion conditions, the maximum flame height is defined as the maximum value of the transient flame height, and the average flame height is defined as the time average of the transient flame height, as shown in Equation (7).

$$h = \frac{\sum_1^n h_i}{n} \tag{7}$$

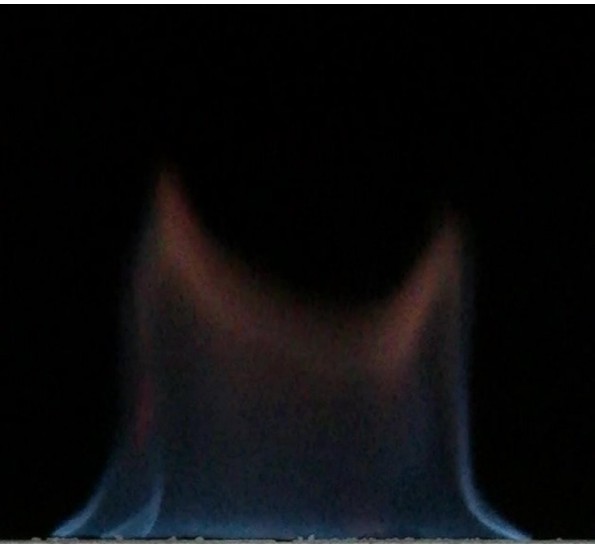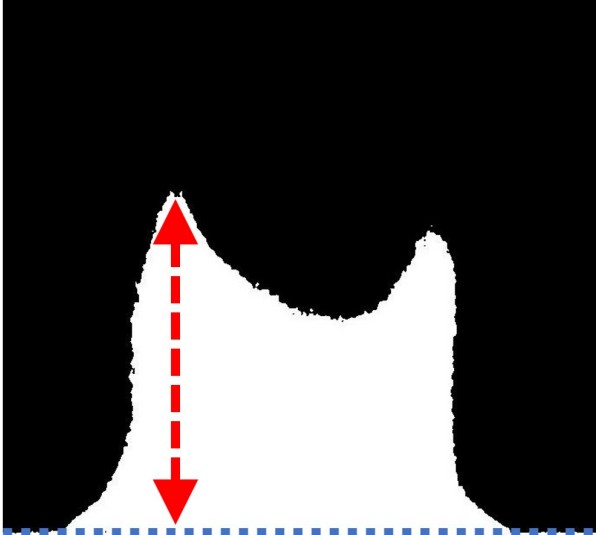

**Figure 2.** Transient image and binarized image of a burning flame.

As the methane hydrate sediment burns, the dissociated water transports and percolates to the bottom of the sample, and the combustion tank absorbs some of the dissociated water and causes the tank to gain weight. Therefore, the discharge water mass is defined as the difference in mass before and after the burning of the combustion tank. In addition, to eliminate the influence of different filling masses on the discharge water mass, the dimensionless discharge water mass is defined as the ratio of discharge water mass to filling mass, and its mathematical expression is shown in Equation (8).

$$\Delta m^* = \frac{m_1 - m_2}{m} \tag{8}$$

where $m_1$ is the mass of the combustion tank before combustion, $m_2$ is the mass of the combustion tank after the end of combustion, and $m$ is the filling mass of the sample.

## 3. Experimental Results and Analysis

### 3.1. Influence Law of Particle Size on the Combustion Characteristics of Methane Hydrate Sediments

Figure 3 shows the images of the flame height of the samples with the variation law of gravel particle size, where Figure 3a shows the trend of the average flame height and Figure 3b shows the trend of the maximum flame height. It can be seen from the figure that the average flame height and the maximum flame height decrease gradually with the increase in particle size. After the flame height reaches the minimum value of 1.5 mm, it increases again. As the particle size decreases, the pore distance between gravel and methane hydrate particles decreases, the heat transfer rate between gravel and methane hydrate particles increases, the amount of methane released by dissociation during combustion increases, and therefore, the flame height increases. However, when the particle size is increased from 1.5 mm to 2 mm, the flame height increases instead of decreasing. The reason for this experimental phenomenon can be explained by the change in the way the methane hydrate particles acquire heat. Methane hydrate sediments produce dissociated water during combustion. The dissociated water absorbs the heat generated by the flame and flows in the pore space, while the unheated gravel and methane hydrate particles can exchange heat with the dissociated water flowing in the pore channel, thus leading to an increase in temperature. Methane hydrate particles have a thermal conductivity of 0.5 W/(m·K) [31,32], while for quartz sand, which has the lowest thermal conductivity, it is 1.35 W/(m·K). Hence, the gravel will preferentially acquire heat. When the gravel particle size decreases, the pore water can exchange heat more rapidly with the gravel and methane hydrate particles, resulting in a larger flame height for small particle

size samples. When the thermal conductivity of the gravel increases, the gravel gains more heat from the pore water and the methane hydrate particles gain less heat, thus the height of the flame decreases. When the particle size increases to 2 mm, the pore distribution is sparse, and the dissociated water in the pore channel cannot effectively exchange heat with the gravel and methane hydrate particles. At this time, the heat exchange in methane hydrate particles is only related to the heat conduction of the gravel, so the flame height can increase again. It can be assumed that the heat transfer process between gravel and methane hydrate particles relies mainly on the transport of dissociated water in the pore channels when the particle size does not exceed 1.5 mm. When the particle size exceeds 1.5 mm, the heat transfer process between the gravel and methane hydrate particles mainly relies on the heat conduction of the gravel inside the sample layer.

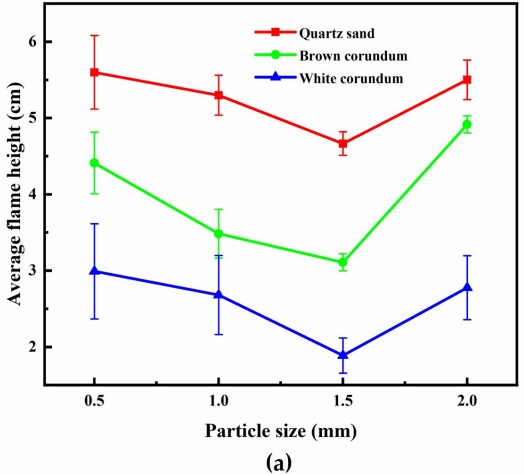
(a)

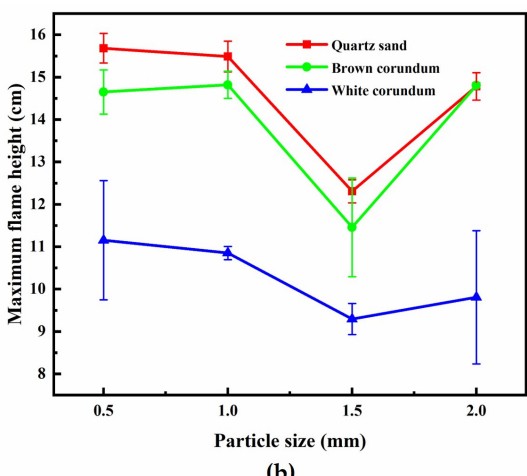
(b)

**Figure 3.** Image of flame height changing with particle size: (**a**) average flame height and (**b**) maximum flame height.

It can also be seen from Figure 3 that as the type of gravel in the methane hydrate sediment samples changes from quartz sand to brown corundum and then to white corundum, the thermal conductivity of the gravel gradually increases and the flame height gradually decreases. When a methane hydrate sediment sample is filled into the combustion tank, convective heat exchange between the surface sample and the air causes the sample temperature to increase, while the lower temperature bottom sample continues to cool the sample to keep the temperature constant. These two heat transfer processes together determine the dissociation state of the methane hydrate sediment sample before ignition. When the thermal conductivity is small, the warming process of the surface sample prevails, and no ice crusts are formed on the surface of the sample before ignition that would hinder the subsequent dissociation and combustion processes. When the thermal conductivity is large, the cooling process of the bottom sample prevails, and the ice crusts are formed on the sample surface before ignition, which in turn leads to a blockage of the subsequent dissociation and combustion processes and a consequent decrease in flame height.

Figure 4 shows the images of the variation law of sample mass with gravel particle size during the combustion process, where Figure 4a shows the trend of effective combustion ratio and Figure 4b shows the trend of combustion ratio per unit time. As can be seen from the figure, with the increase in particle size, the effective combustion ratio first gradually decreases and then gradually increases, but the change is not obvious. Meanwhile, with the increase in particle size, the combustion ratio per unit time gradually decreases, reaches its minimum value at 1.5 mm, and then increases again. The particle size increases, the pore distribution becomes sparse, and the heat transfer of dissociated water in the pore channels becomes more impeded, resulting in a decrease in the heat transfer rate of methane hydrate particles and hence the combustion ratio per unit time. When the particle size is 1.5 mm,

the superposition of the transport heat transfer process of the pore channel dissociated water and the heat transfer process of the gravel particle size is the least effective, resulting in the average flame height, maximum flame height, and combustion ratio per unit time of the particle size being the smallest among all the particle size ranges. In addition, samples with different particle sizes have similar effective combustion ratios, which can indicate that the samples can finalize the heat transfer process, either by transporting heat through dissociated water within the pore channels or by direct heat transfer through the gravel. The flame height is related to the heat released rate [33,34], which is directly related to the dissociation and combustion of methane hydrate particles, so the two heat transfer processes affect the rate of heat acquisition by methane hydrate particles during sample combustion rather than the amount of methane hydrate particles dissociated.

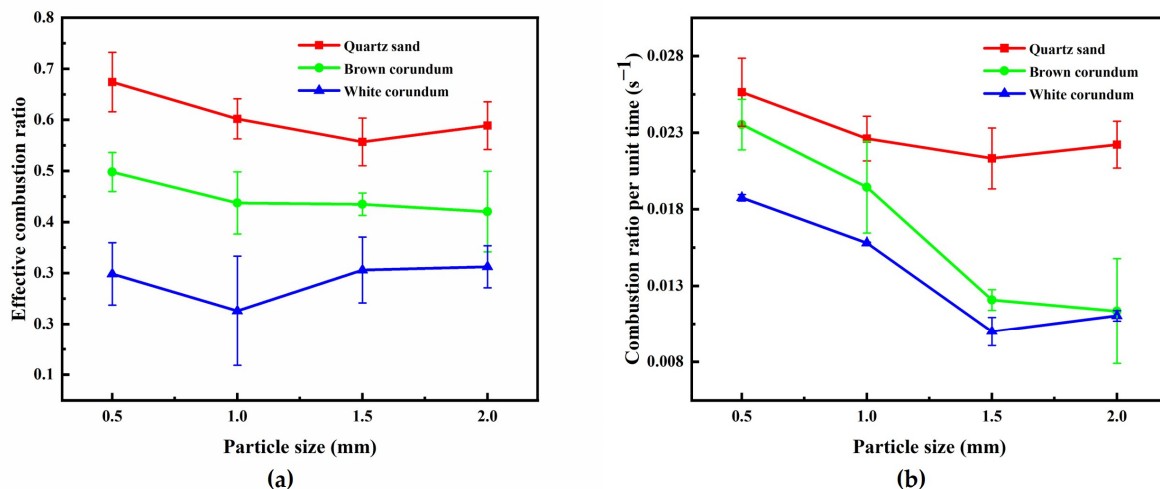

**Figure 4.** Image of sample mass changing with particle size: (**a**) effective combustion ratio and (**b**) combustion ratio per unit time.

In addition, it can be observed from the figure that with the increase in thermal conductivity, the effective combustion ratio and combustion ratio per unit time are significantly reduced. When the thermal conductivity is small (quartz sand), with the increase in particle size, the change in combustion ratio per unit time is not significant. When the thermal conductivity is large (brown corundum and white corundum), with the increase in particle size, the magnitude of the change in combustion ratio per unit time is obvious. When the thermal conductivity increases, the dissociated water formed by combustion is cooled by the substrate cryogenic sample, and ice crusts are formed, which hinder the release of subsequent dissociated gases and increase the heat dissipation of the flame, thus leading to a decrease in the effective combustion ratio and the combustion ratio per unit time.

### 3.2. Influence Law of Porosity on Combustion Characteristics of Methane Hydrate Sediments

In order to study the influence law of the variable of porosity on the combustion characteristics of methane hydrate sediments, the particle size of the samples is controlled at 1 mm, and the saturation of the samples is controlled at 0.7. Figure 5 shows the variation law of flame height with porosity, where Figure 5a shows the trend of average flame height and Figure 5b shows the trend of maximum flame height. As can be seen from the figure, with the increase in porosity, the average height of the flame and the maximum height of the flame both gradually increase. The larger the porosity is, the larger the volume of pores is, and the hindrance to the seepage process of dissociation water is reduced. At the same time, the heat and mass transfer obstacles between the sample and the flame are reduced, and it is less likely to form an ice layer that limits the combustion effect due to the self-protection effect during combustion, so the flame height increases. In addition, the increase in flame height is also related to the increase in the number of methane hydrate

particles. Since the saturation is controlled to be 0.7 in the experiment, which means the ratio of methane hydrate particles to pore volume is 0.7, the pore volume will gradually increase as the porosity increases, and the number of methane hydrate particles will also increase to ensure the relative ratio remains the same.

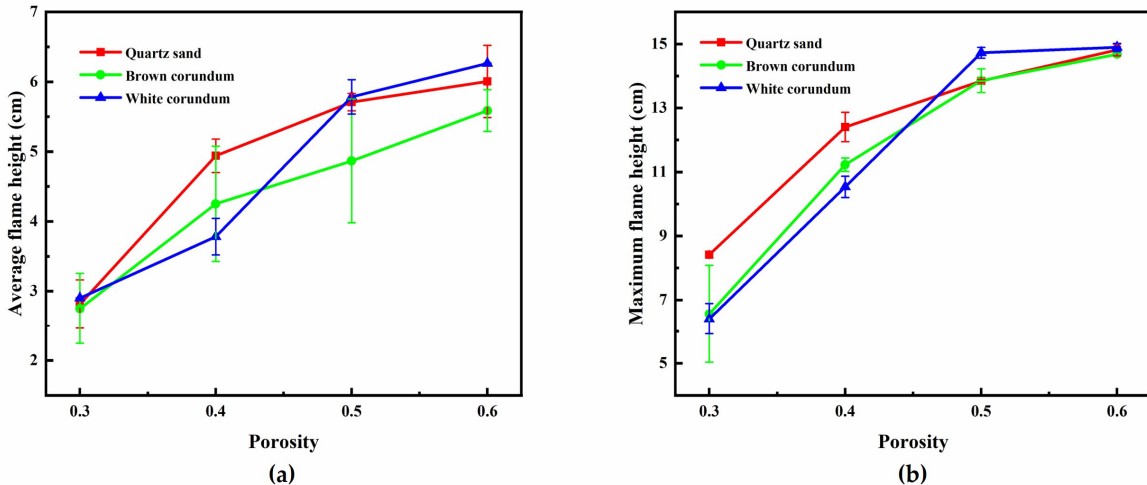

**Figure 5.** Image of flame height changing with porosity: (**a**) average flame height and (**b**) maximum flame height.

It can also be seen from the figure that at low porosity (0.3 and 0.4), both the average flame height and the maximum flame height decrease gradually as the thermal conductivity increases. While at high porosity (0.5 and 0.6), as the thermal conductivity increases, the average flame height first decreases and then increases, and the maximum flame height is almost constant or slightly increases. As mentioned earlier, methane hydrate particles can acquire heat through the transport heat transfer process of dissociated water in the pore channels and the heat transfer process of gravels. When the porosity is small, the volume occupied by pores is relatively small, and the volume occupied by gravel is relatively large, so the methane hydrate particles can only obtain heat through the heat conduction process of gravel. Moreover, the larger the thermal conductivity, the more preferentially the gravel acquires heat, and the more difficult it is for the methane hydrate particles to acquire heat, so the white corundum sample with the largest thermal conductivity has the smallest flame height. When the porosity is large, the volume occupied by the pores is relatively large, while the volume occupied by the gravel is relatively small. At this time, the methane hydrate particles can obtain heat through the transport heat transfer process of dissociated water and the heat conduction process of gravel at the same time, and the flame height of the brown corundum sample is the smallest under the superposition of the two heat transfer processes.

Figure 6 shows the images of the variation law of sample mass with porosity during combustion, where Figure 6a shows the trend of effective combustion ratio and Figure 6b shows the trend of combustion ratio per unit time. It can be seen from Figure 6a that when the thermal conductivity is small (quartz sand and brown corundum), with the increase in porosity, the effective combustion ratio gradually increases, reaches its maximum value at 0.5, and then gradually decreases. When the thermal conductivity is large (white corundum), the effective combustion ratio gradually increases with the increase in porosity. As can be seen from Figure 6b, with the increase in porosity, the combustion ratio per unit time gradually increases, and when the porosity exceeds 0.5, it begins to decrease, while the combustion ratio per unit time gradually decreases with the increase in thermal conductivity. When the porosity is small, the heat transfer rate of methane hydrate particles is influenced by the thermal conductivity, so the effective combustion ratio and the combustion ratio per unit time both decrease with the increase in thermal conductivity. When the porosity is relatively large, the number of methane hydrate particles increases consequently, so the

sample needs to absorb more heat to decompose methane hydrate during combustion. When the transport heat transfer process of dissociated water in the pore channel and the heat conduction process of gravel cannot provide the heat required for dissociation in time, the combustion ratio and combustion ratio per unit time are reduced as a result.

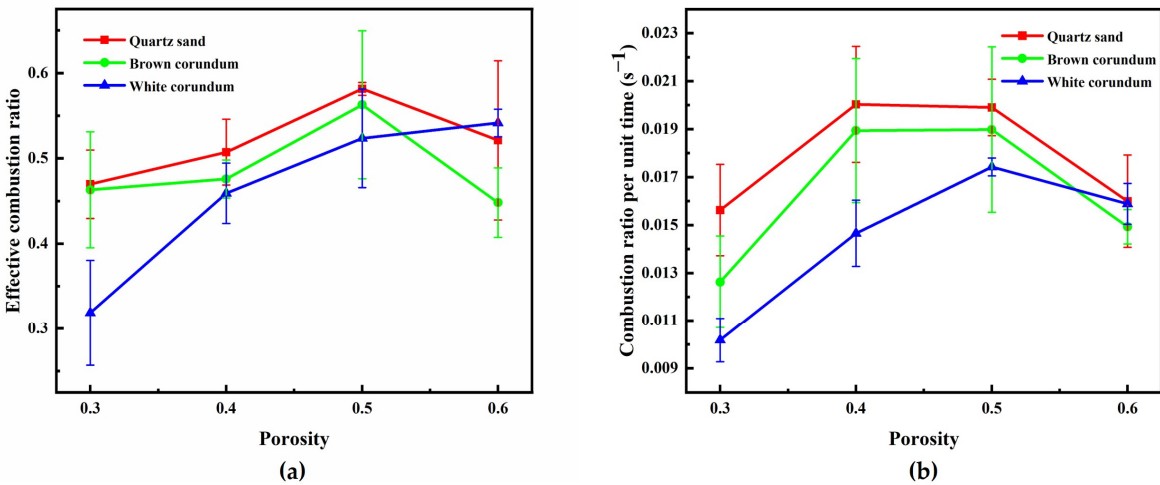

**Figure 6.** Image of sample mass changing with porosity: (**a**) effective combustion ratio and (**b**) combustion ratio per unit time.

Figure 7 shows the variation law of dimensionless discharge water mass with porosity. It can be seen from the figure that the dimensionless discharge water mass gradually increases with the increase in porosity. Moreover, it can also be observed from the figure that the dimensionless discharge water mass decreases as the thermal conductivity increases. It can be speculated that the dissociated water in the pore channels also increases with the increase in porosity. As the porosity increases, the volume occupied by the pores increases, the obstruction to the seepage of dissociated water in the pore channels decreases, and more dissociated water can be transported to the bottom of the sample layer and absorbed by the combustion tank. Therefore, the discharge water mass increases with the increase in porosity. From this, it is clear that to improve the in-situ combustion of methane hydrate sediments, increasing the porosity of the samples would be an effective combustion optimization measure.

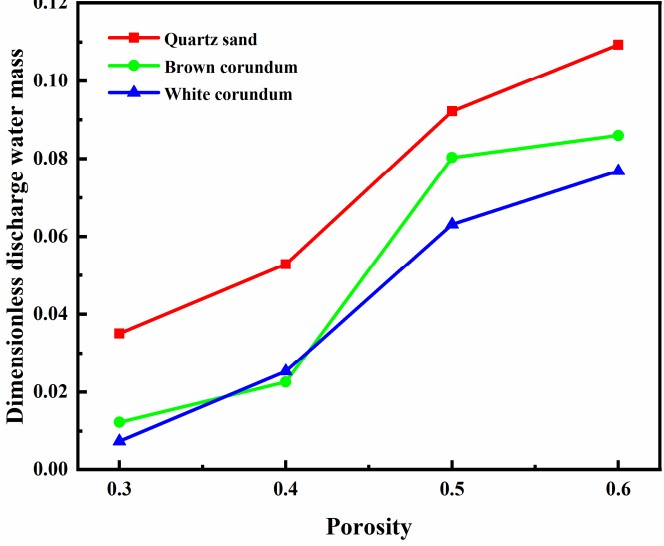

**Figure 7.** Image of dimensionless discharge water mass changing with porosity.

### 3.3. Influence Law of Saturation on Combustion Characteristics of Methane Hydrate Sediments

In order to investigate the effect law of the variable of saturation on the combustion characteristics of methane hydrate sediments, the particle size of the samples is controlled at 1 mm and the porosity of the samples is controlled at 0.4. Since the total volume, particle size, and porosity of the sample remain constant, when saturation changes, the total pore volume and the volume occupied by gravel remain unchanged, while the volume occupied by methane hydrate particles and the proportion of methane hydrate particles to pore volume change. Figure 8 shows the variation law of flame height with saturation, where Figure 8a shows the trend of average flame height and Figure 8b shows the trend of maximum flame height. It can be seen from the figure that the average flame height and the maximum flame height both increase gradually with the increase in saturation, and with the increase in thermal conductivity, the average flame height and maximum flame height decrease gradually. For the samples formed by mixing with quartz sand or brown corundum, when the saturation exceeds 0.6, the increase in saturation does not make the maximum flame height increase significantly. For the sample formed by mixing with white corundum, when the saturation exceeds 0.6, the increase in saturation still causes the maximum flame height to increase significantly. When the saturation increases, the number of methane hydrate particles and the proportion of pore volume they occupy increase, and more dissociated methane will participate in the reaction during combustion, thus increasing the flame height. When the saturation exceeds 0.6, the maximum flame height of the sample formed by mixing quartz sand or brown corundum does not increase with the increase in saturation, indicating that the intensity of the combustion reaction is controlled by the concentration of oxidant at this time, and the insufficient concentration of oxidant limits the continued increase in maximum flame height. For the sample formed by mixing with white corundum, when the saturation exceeds 0.6, the flame extremum height still increases with the increase in saturation, which indicates that the dissociation rate of methane hydrate particles has not reached its maximum at this time, and the flame thermal dissipation caused by the larger thermal conductivity limits the heat transfer process of methane hydrate particles. In contrast, a smaller thermal conductivity reduces the thermal dissipation of the flame, and more heat is available to the methane hydrate particles, so the sample of quartz sand with the smallest thermal conductivity has the largest flame height.

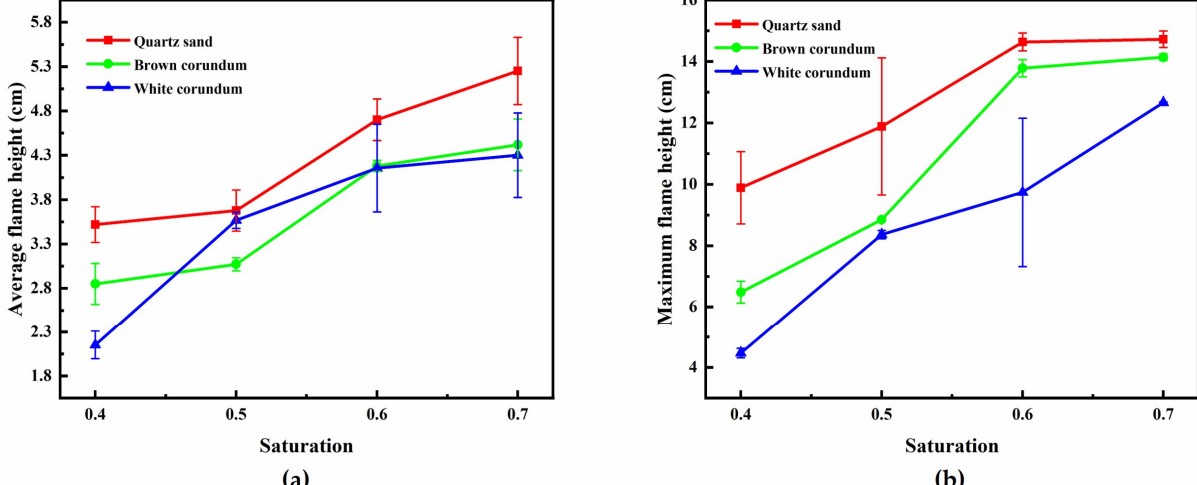

**Figure 8.** Image of flame height changing with saturation: (**a**) average flame height and (**b**) maximum flame height.

Figure 9 shows the images of the variation law of sample mass with saturation during combustion, where Figure 9a shows the trend of the effective combustion ratio and Figure 9b shows the trend of the combustion ratio per unit time. As can be seen from the figure, with the increase in saturation, the effective combustion ratio and combustion ratio per unit

time both gradually increase, and with the increase in thermal conductivity, the effective combustion ratio and combustion per unit time both gradually decrease. For the samples formed by mixing quartz sand, when the saturation increases, the effective combustion ratio changes significantly, while the combustion ratio per unit time changes insignificantly. For the samples formed by mixing brown or white corundum, when the saturation increases, the effective combustion ratio changes insignificantly, while the combustion ratio per unit time changes significantly. When the saturation increases, the number of methane hydrate particles increases, the amount of methane involved in the reaction during sample combustion increases, and the intensity of the combustion reaction and flame height are enhanced, which further promotes the dissociation of methane hydrate particles, and the effective combustion ratio increases as a result. Comparing Figures 8 and 9, it can be found that for the samples formed by the quartz sand mixture, the intensity of the combustion reaction is significantly enhanced due to the increase in saturation, so that all three combustion characteristics parameters, that is, the average flame height, maximum flame height, and effective combustion ratio, change significantly with the change in saturation.

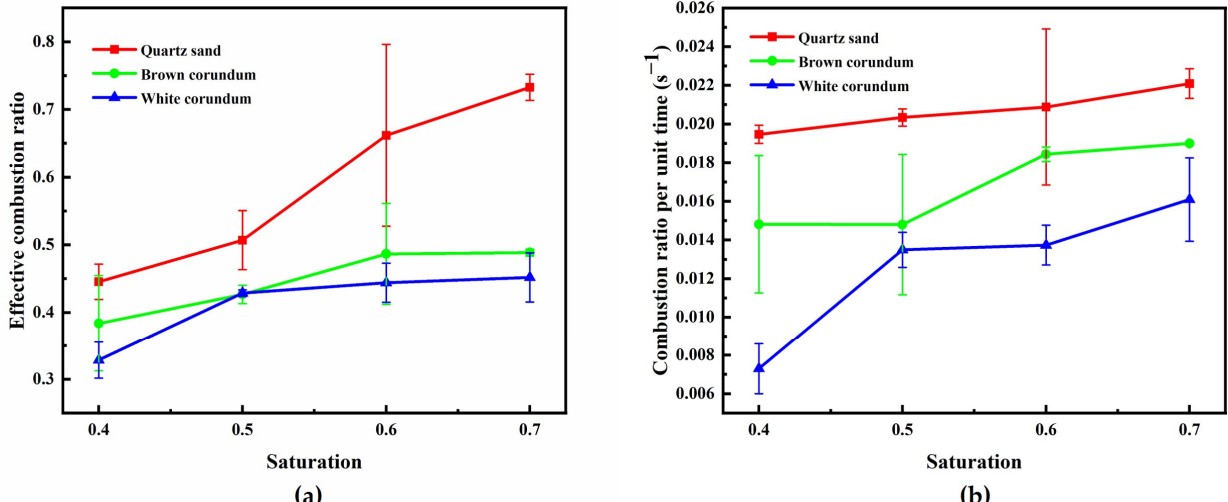

**Figure 9.** Image of sample mass changing with saturation: (**a**) effective combustion ratio and (**b**) combustion ratio per unit time.

Figure 10 shows the variation law of dimensionless discharge water mass with saturation. It can be seen from the figure that when the saturation does not exceed 0.6, the dimensionless discharge water mass first decreases or remains basically the same as the saturation increases. When the saturation exceeds 0.6, the dimensionless discharge water mass increases sharply with the increase in saturation. It can also be observed from the figure that as the thermal conductivity increases, the dimensionless discharge water mass gradually decreases. When the saturation is 0.7, the discharge water mass of samples with different thermal conductivity reaches its maximum value, which is rooted in the reduction of the heat and mass transfer resistance between methane hydrate particles and the flame, which increases the amount of methane involved in the combustion reaction and thus leads to the improvement of the combustion of the samples. Therefore, to improve the in-situ combustion of methane hydrate sediments, the use of high-saturation in-situ sediment samples would be an effective optimization measure. In addition, as mentioned earlier, the oxidant concentration at high saturation limits the increase in the intensity of the combustion reaction. Therefore, increasing the oxidant concentration and ensuring the increase in reaction intensity is a reliable optimization solution when using high saturation samples for combustion.

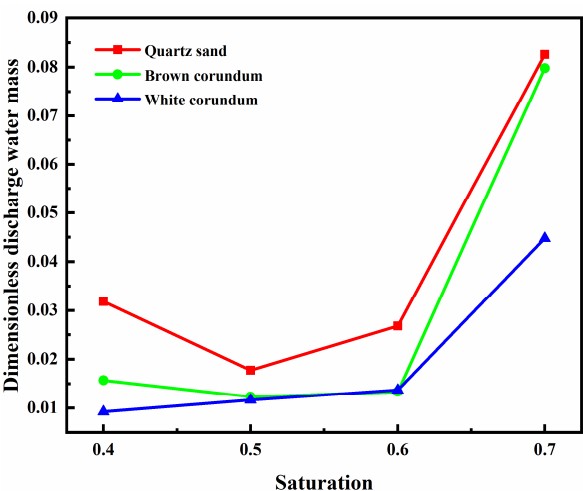

**Figure 10.** Image of dimensionless discharge water mass changing with saturation.

## 4. Numerical Calculation and Discussion of Results

### 4.1. Model Building

First, the combustion process of methane hydrate sediments needs to be further simplified. The essence of the methane hydrate sediment combustion phenomenon is the release of methane by the dissociation of methane hydrate particles and the combustion reaction of methane with an oxidizer to produce a flame [35]. Therefore, the methane hydrate particles can be simplified to a circular methane velocity inlet, and the methane released rate is obtained by experimental measurement. In addition, the methane hydrate sediment sample in the combustion tank is simplified to a mixture formed by methane hydrate particles and gravel, with round methane hydrate particles layered with round gravel and spaced inside the sample layer. According to the experimental measurement results, the maximum transient mass change in the sample is 0.3 g/s, and then the methane inlet velocity can be calculated as $2.983 \times 10^{-5}$ m/s based on the methane density and the cross-sectional area of the combustion tank.

The overall schematic diagram of the model boundary conditions is shown in Figure 11, the main body of which can be divided into the Pore area and the Outer area. The Pore area is the porous media area of the methane hydrate sediments in the combustion tank, where the methane hydrate particle spheres release methane, which flows in the pore channels within the porous media area and reaches the Outer area to participate in the reaction. The overall size of the Pore area is set to 50 mm × 10 mm with reference to the cross-sectional dimensions of the combustion tank. The main boundary conditions contained in the Pore area are the methane velocity inlet Inlet, the gravel wall Wall1, the combustion tank bottom wall Wall2, the combustion tank side wall Wall3, the symmetry axis Sym1, and the grid intersection Interface1. Since both the methane velocity inlet and the gravel wall are circular, O-grid is used to mesh the Pore area to improve the grid quality. The Outer area is the reaction area between dissociated methane and oxidant (air), and its overall size is set at 120 mm × 390 mm. In this area, dissociated methane flowing from the Pore area will undergo heat and mass transfer, momentum transfer, and other processes with air and undergo chemical reactions. The Outer area contains the main boundary conditions of the air outlet Outlet, the symmetry axis Sym2, and the grid intersection Interface2. Since the Outer area is a regular rectangle, an unstructured grid is used for meshing the Outer area. In the entire Outer area, some areas are chemical reaction areas that need to be encrypted, and some areas are only airflow areas that do not need to be encrypted. Therefore, the distance of grid nodes is adjusted to achieve sparse and encrypted grids during unstructured meshing, while the variation ratio of node distance is set to 1.3 to ensure the quality of meshing.

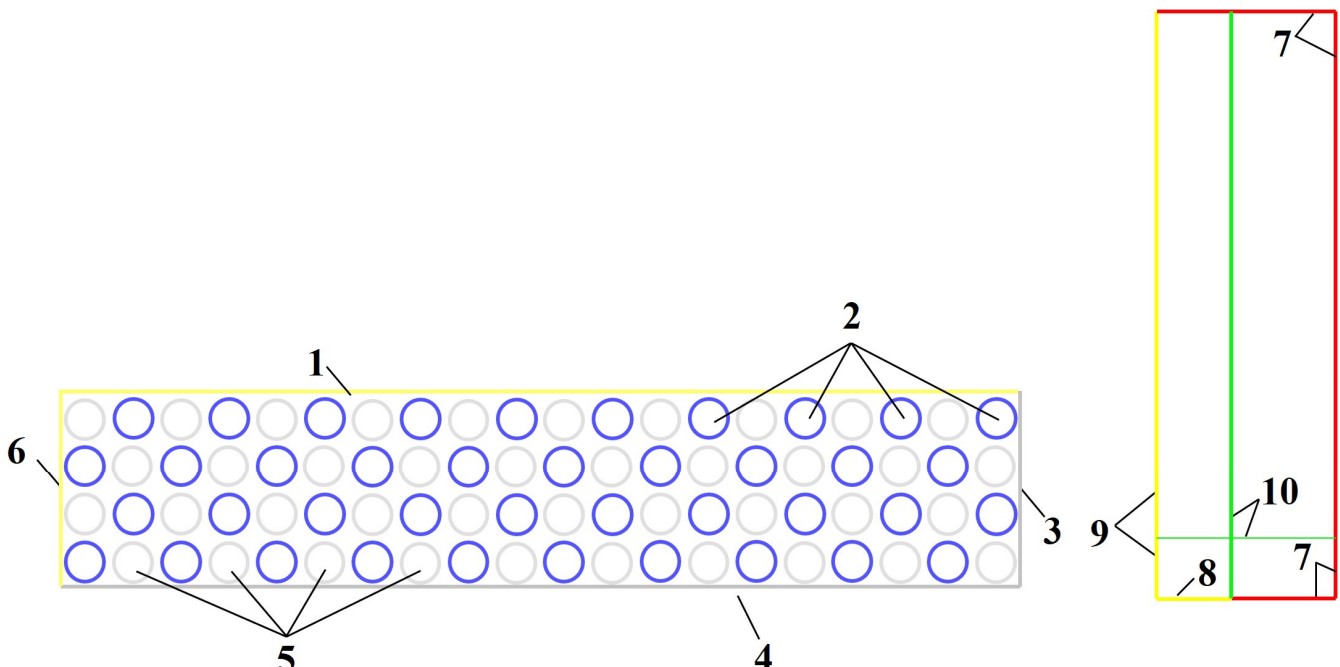

**Figure 11.** Schematic diagram of the boundary conditions: 1—Interface1, 2—Inlet, 3—Wall3, 4—Wall2; 5—Wall1, 6—Sym1, 7—Outlet, 8—Interface2, 9—Sym2, and 10—Interior.

Dagan et al. [36] assumed that during the combustion process, methane hydrate migrates in the form of monodisperse particles. Building upon the aforementioned spherical particle combustion model, they further constructed a laminar spray combustion model. This model provided parameters such as component concentrations, flame temperature, and flame propagation rate. It was demonstrated that this theoretical model, in comparison to the $D^2$ law [37], aligns more closely with reality. However, it is worth noting that the model idealizes the diffusion and mixing processes of gases, and the described combustion flame is simplified as a one-dimensional flame.

Bar-Kohany et al. [36,38] developed a three-phase structure model for methane hydrate spheres (as illustrated in Figure 12). This three-phase structure comprises a solid hydrate core, liquid water, a bubbly mixture of gaseous methane, and a convective diffusive steady-state gas phase. Using this model, Bar-Kohany et al. investigated the combustion characteristics of methane hydrate under different environmental conditions, including temperature, pressure, composition, and the methane-to-water mass ratio. Their findings revealed that the driving force behind the evaporation and combustion processes stems from the heat flux at the droplet surface caused by heat exchange between the flame and ambient hot air. A decrease in ambient temperature and a reduction in fuel content in the hydrate result in a decrease in fuel mass flow, evaporation rate of water, and fuel combustion rate. A reduction in methane content or an increase in water content leads to a decrease in flame temperature, subsequently causing a reduction in flame height and hydrate surface temperature.

From the comprehensive literature review, it is evident that the referenced papers have modeled data without the integration of actual methane combustion processes. Furthermore, they primarily focus on pure hydrates and do not take into account lithological parameters. In contrast, this paper concentrates on the study of the combustion characteristics of methane hydrate sediment under various lithological parameters. The model employed in this study is concise and closely aligns with experimental data, making it suitable for computational analysis. However, it is important to note that the numerical simulations in this paper are limited to steady-state calculations and do not consider the complex multiphase flow within the sediment. Additionally, when considering simplifica-



tions in terms of inlet velocity and porous region, it may be worthwhile to introduce more complexity factors for a comprehensive investigation.

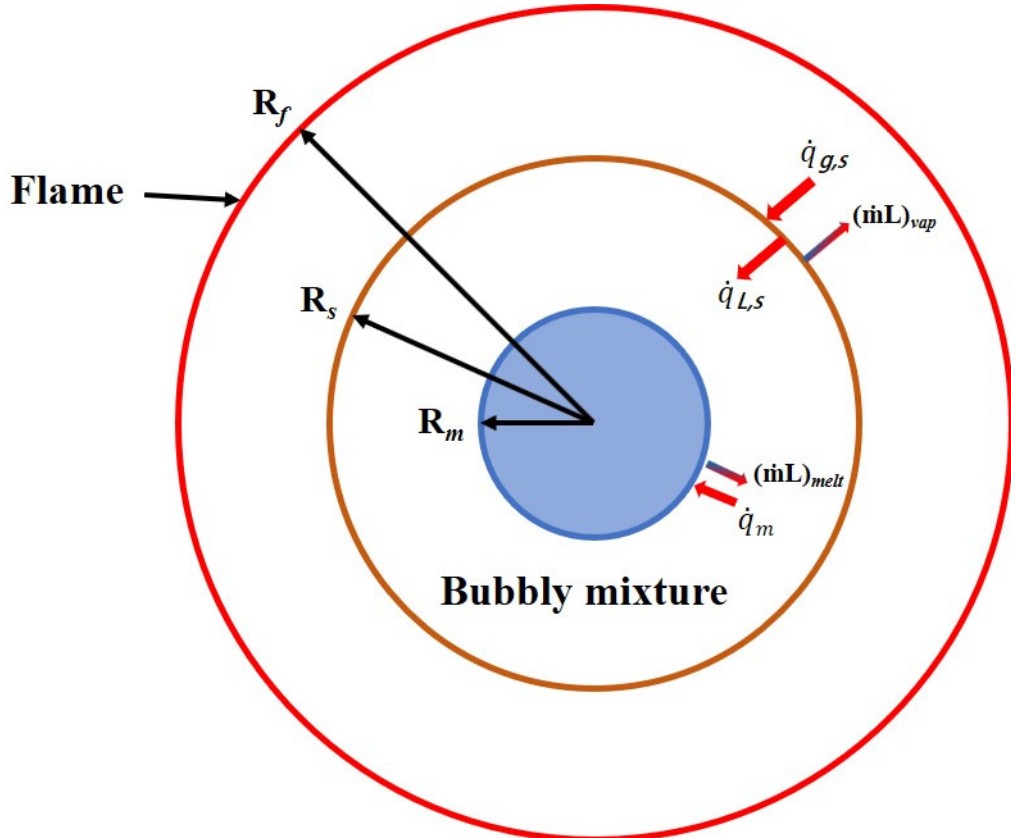

**Figure 12.** Three-phase structure model of methane hydrate [38].

*4.2. Parameter Settings and Boundary Conditions*

Once the grid model is established, it needs to be imported into FLUENT, and the following conditions should be set to complete the simulation calculations:

(1) General Settings: Choose the pressure-based solver, opt for steady-state calculations, and set the gravitational acceleration in the Y direction to $-9.81$ m/s$^2$.

(2) Model Settings: Activate the energy equation, choose the RNG model within the k-epsilon turbulence model in the viscous model settings, select the standard wall functions, implement the species transport model and enable volume reactions, choose the methane–air mixture as the composite material, and select the Eddy-Dissipation model for the interaction between turbulence and chemical reactions.

(3) Boundary Conditions: Set the outlet boundary condition as a constant pressure outlet with an outlet temperature of 300 K and an oxygen mass fraction of 0.23; define the inlet boundary condition as a velocity inlet, calculate the velocity magnitude based on the experimentally determined sample mass change curve, and set the inlet temperature to 263 K and a methane mass fraction of 1.

(4) Solution Methods: Choose the SIMPLE algorithm as the solution method, select the Green–Gauss Node-Based scheme for gradient discretization, and use Second Order Upwind for the discretization of other parameters.

(5) Residual Settings: Set the energy residual to $10^{-6}$, and set all other residuals to $10^{-4}$.

(6) Local Initialization Settings: Set the temperature in the Outer region to 300 K and the temperature in the Pore region to 263 K, assign an oxygen mass fraction of 0.23 to the Outer region and 0 to the Pore region, and in both the Outer and Pore regions, initialize the velocity, methane mass fraction, and water mass fraction to 0.

(7)  In order to simulate the combustion process of methane hydrate sediments, it is necessary to obtain the mass change curve of methane hydrate sediments through the combustion experiment and then obtain the methane inlet velocity required by the simulation calculation. The maximum instantaneous mass change in the sample is 0.3 g/s, and the methane inlet velocity is $2.983 \times 10^{-5}$ m/s, according to the methane density and the cross-sectional area of the combustion chamber.

For the same physical model size, dividing the grid into more cells results in smaller cell sizes, which leads to higher grid accuracy and computational precision. However, this also consumes more computational resources and increases the simulation time. To perform grid independence validation, grid models with minimum grid sizes of 0.5 mm, 0.25 mm, 0.1 mm, and 0.0625 mm have been created. The minimum grid size is defined as the smallest size of the Pore region's O-grid mesh, which represents the area where methane flows out. The grid accuracy in this area significantly affects the simulation results. The grid size in the Outer region is scaled up proportionally from the minimum grid size with a scaling factor of 1.3. Here are the total grid cell counts for the four different grid sizes: 0.5 mm minimum size: 55,553 total grid cells, 0.25 mm minimum size: 159,587 total grid cells, 0.1 mm minimum size: 320,212 total grid cells, and 0.0625 mm minimum size: 903,752 total grid cells.

The gas's maximum velocity obtained from the simulation for the four grid sizes is shown in Figure 13. From the figure, it is evident that as the minimum grid size decreases, the maximum gas velocity obtained from the simulation gradually decreases. This is because reducing the grid size makes each iteration more precise, resulting in simulation results that closely approximate the actual combustion conditions. Based on the figure, it is apparent that using a grid model with a minimum grid size of 0.1 mm is the most suitable choice for simulation calculations.

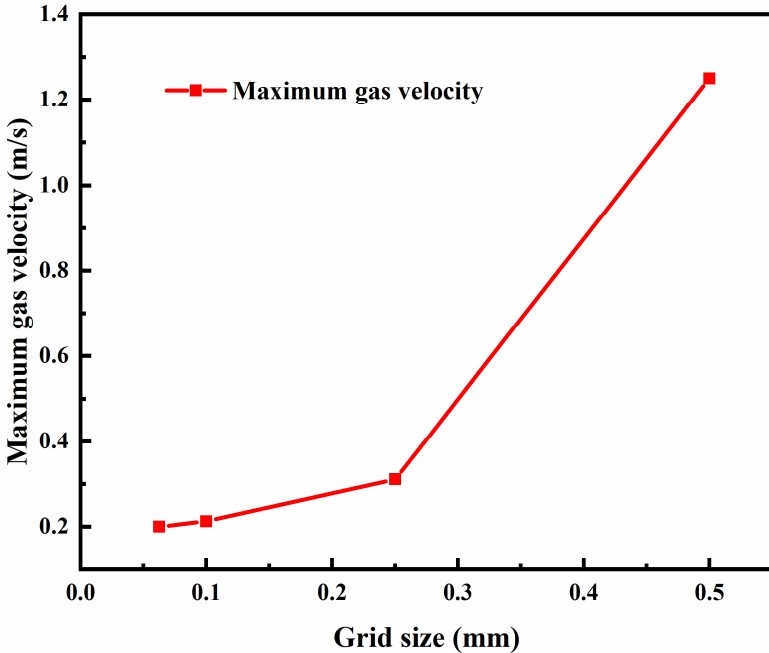

**Figure 13.** Grid independence verification.

*4.3. Governing Equations for Numerical Computing*

The control equations for the combustion process are as follows:

$$p = \frac{\rho RT}{M} \tag{9}$$

$$\frac{\partial(\rho u)}{\partial x} + \frac{\partial(\rho v)}{\partial y} = 0 \tag{10}$$

$$\frac{\partial(\rho u^2)}{\partial x} + \frac{\partial(\rho uv)}{\partial y} = \frac{\partial \tau_{xx}}{\partial x} + \frac{\partial \tau_{yx}}{\partial y} - \frac{\partial p}{\partial x} \tag{11}$$

$$\frac{\partial(\rho v^2)}{\partial y} + \frac{\partial(\rho uv)}{\partial x} = \frac{\partial \tau_{yy}}{\partial y} + \frac{\partial \tau_{xy}}{\partial x} - \frac{\partial p}{\partial y} \tag{12}$$

$$\frac{\partial[u(\rho E + p)]}{\partial x} + \frac{\partial[v(\rho E + p)]}{\partial y} = \frac{\partial(u\tau_{xx} + v\tau_{yy} - q_x)}{\partial x} + \frac{\partial(v\tau_{yy} + u\tau_{yx} - q_y)}{\partial y} + q\dot{w} \tag{13}$$

$$\frac{\partial(\rho u\overline{Y})}{\partial x} + \frac{\partial(\rho v\overline{Y})}{\partial y} = \frac{\partial}{\partial x}(\rho D \frac{\partial \overline{Y}}{\partial x}) + \frac{\partial}{\partial y}(\rho D \frac{\partial \overline{Y}}{\partial y}) - \dot{w} \tag{14}$$

$$\tau_{xx} = \frac{4}{3}\mu \frac{\partial u}{\partial x} - \frac{2}{3}\mu \frac{\partial v}{\partial y} \tag{15}$$

$$\tau_{yy} = \frac{4}{3}\mu \frac{\partial v}{\partial y} - \frac{2}{3}\mu \frac{\partial u}{\partial x} \tag{16}$$

$$\tau_{xy} = \tau_{yx} = \mu \frac{\partial u}{\partial y} + \mu \frac{\partial v}{\partial x} \tag{17}$$

$$q_x = -K\frac{\partial T}{\partial x}, q_y = -K\frac{\partial T}{\partial y} \tag{18}$$

$$E = \frac{p}{(\gamma - 1)\rho} + \frac{1}{2}\left(u^2 + v^2\right) \tag{19}$$

$$k = K/\left(qc_p\right) \tag{20}$$

$$c_p = \gamma R/(\gamma - 1) \tag{21}$$

$$\mu = \mu_0 T^n, D = D_0 T^n, k = k_0 T^n \tag{22}$$

$$\dot{w} = d\overline{Y}/dt = A\rho\overline{Y}exp(\frac{-E_a}{RT}) \tag{23}$$

where $A$ is the pre-exponential factor, mol·cm$^{-3}$·s$^{-1}$; $c_p$ is the constant-pressure specific heat, J·Kg$^{-1}$·K$^{-1}$; $D$ is the mass diffusivity, m$^2$·s$^{-1}$; $E$ is the total energy, J·Kg$^{-1}$; $E_a$ is the activation energy, kcal·mol$^{-1}$; $k$ is the thermal diffusivity, m$^2$·s$^{-1}$; $M$ is the molar mass, kg·mol$^{-1}$; $n$ is the constant, 0.7; $p$ is the pressure, Pa; $q$ is the heat release, J·m$^{-3}$·s$^{-1}$; $q_x$ and $q_y$ are the heat fluxs, w·m$^{-2}$; $R$ is the gas constant, J·mol$^{-1}$·K$^{-1}$; $t$ is the time, s; $T$ is the temperature, K; $u$ is the horizontal velocity, m·s$^{-1}$; $v$ is the vertical velocity, m·s$^{-1}$; $x$ is the abscissa; $y$ is the ordinate; $\overline{Y}$ is the mass fraction; $\tau_{xx}$, $\tau_{yy,}$ and $\tau_{xy}$ are the stress tensor, Pa; $\mu$ is the viscosity, Pa·s; $\omega$ is the reaction rate, kg·m$^{-3}$·s$^{-1}$; $\rho$ is the density, kg·m$^{-3}$; and $\gamma$ is the specific heat capacities, J·Kg$^{-1}$·K$^{-1}$.

### 4.4. Mechanism of Particle Size Effect on the Combustion Process of Methane Hydrate Sediments

Figure 14 shows the temperature field distribution of methane hydrate sediments at different particle sizes. From the figure, the area of the high-temperature reaction zone with a temperature over 800 K gradually decreases as the particle size increases. The simulation results in Figure 14 are consistent with the analysis in Section 3.1, that is, a

smaller particle size sample will consume more methane when reacting, and the area of the high-temperature reaction zone will therefore increase, leading to a direct transition of the combustion evolution stage from the intense combustion stage to the flame recession stage.

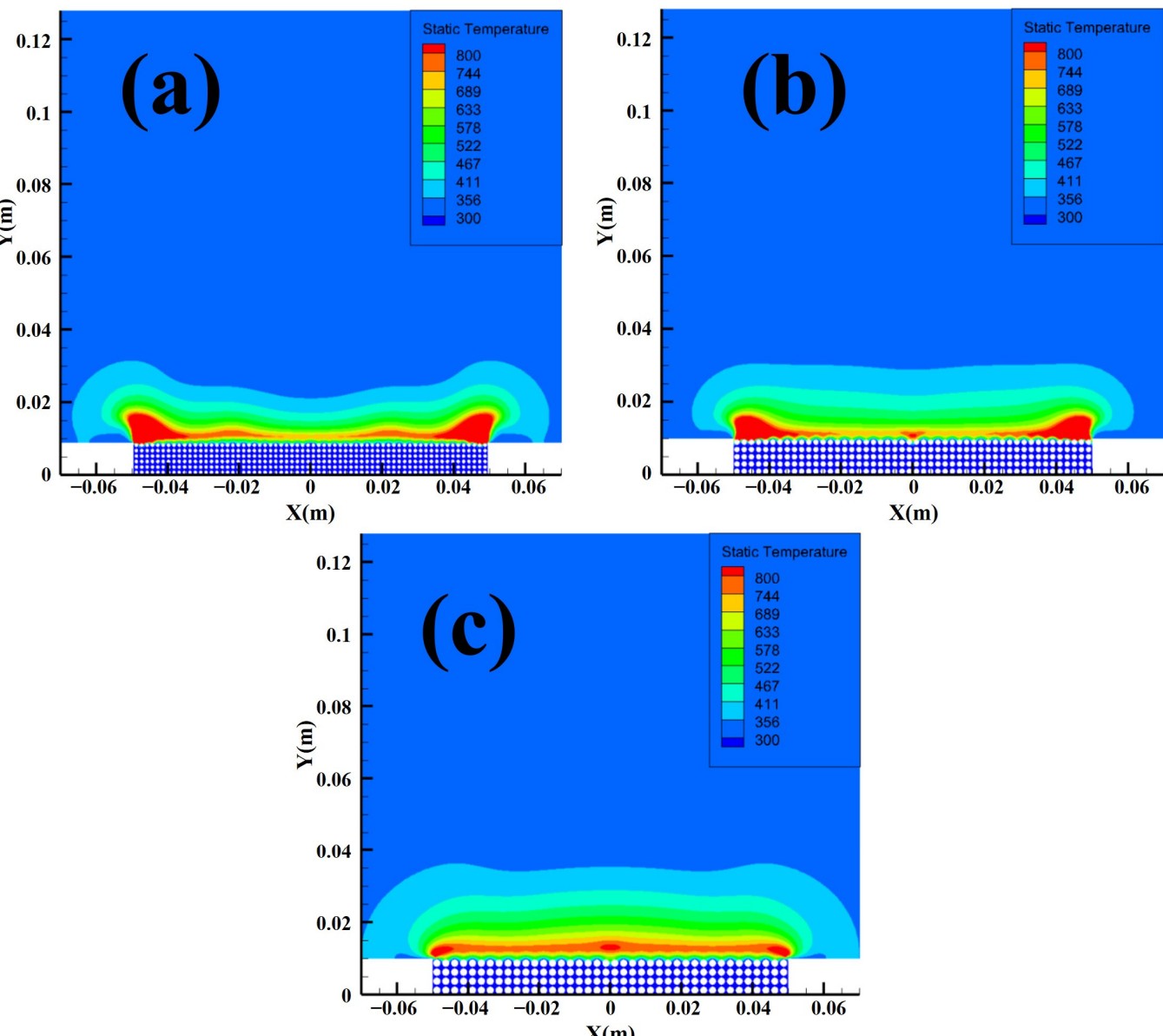

**Figure 14.** Temperature distribution of methane hydrate sediments with different particle sizes: (**a**) 1 mm, (**b**) 1.5 mm, and (**c**) 2 mm.

Since the methane velocity inlets are all located in the Pore area, it is necessary to focus on the concentration distribution of methane in the pore channels in the Pore area. Figure 15 shows the methane concentration distribution in the local area of Pore with different particle sizes. It can be seen from the figure that the 1.5 mm grid model has the lowest methane concentration in the Pore area compared to the 1 mm and 2 mm grid models, while the heat transfer between methane hydrate particles and gravel is the weakest. The simulation results in Figure 15 match the variation law in Figure 3, that is, when the gravel particle size increases from 1 mm to 1.5 mm, the heating effect of the transport heat transfer process of dissociated water for methane hydrate particles is significantly weakened, and the amount of methane released from sample dissociation decreases, which in turn leads to a decrease

in the average flame height and maximum flame height. When the gravel particle size increases from 1.5 mm to 2 mm, the methane hydrate particles can obtain heat through the heat conduction process of the gravel, the amount of methane released by dissociation of the sample increases again, and the average height of the flame and the maximum height of the flame increase again as a result.

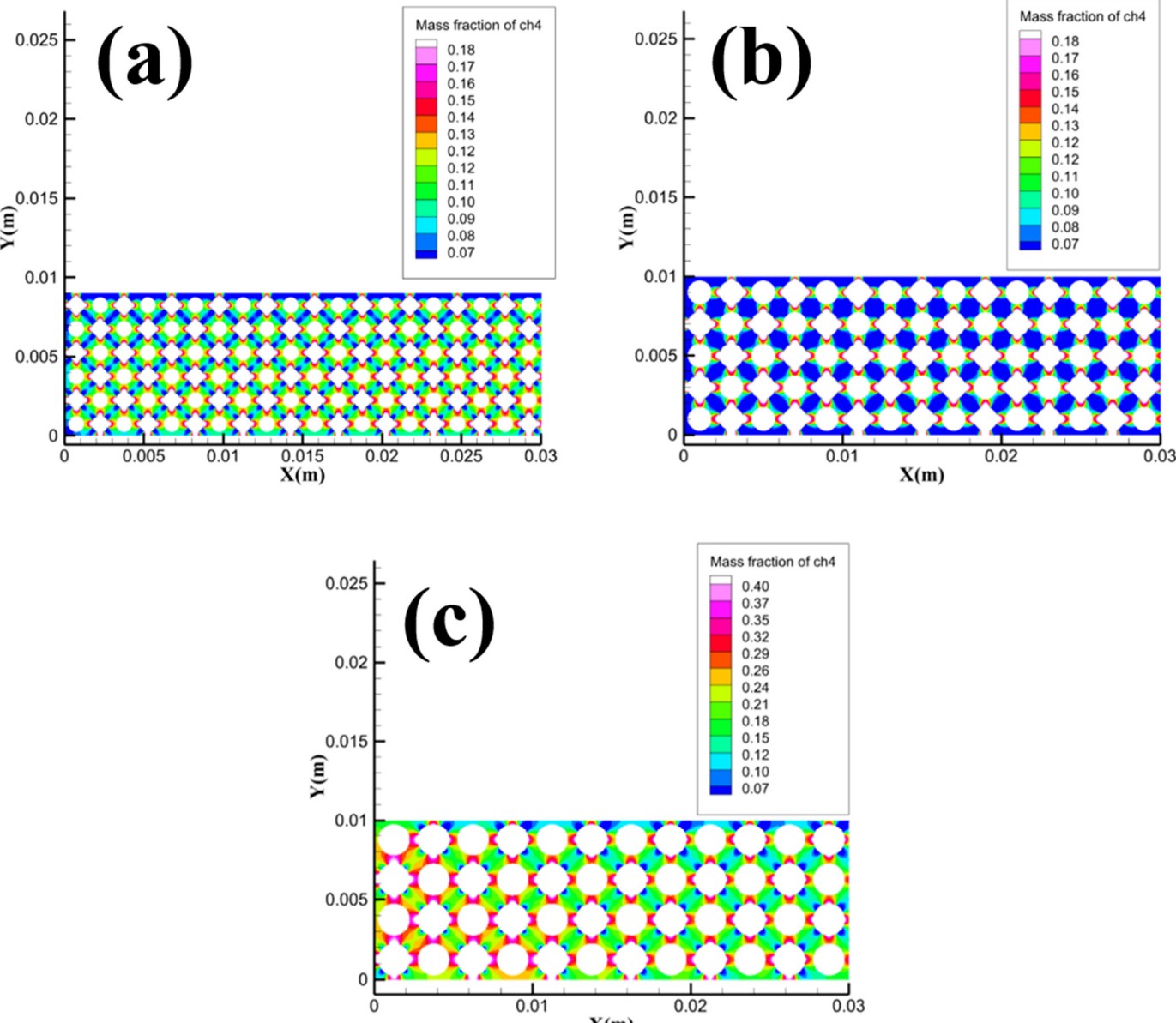

**Figure 15.** Concentration distribution of methane in the local area of Pore under different particle sizes: (**a**) 1 mm, (**b**) 1.5 mm, and (**c**) 2 mm.

Dissociated water within the pore channels is very important for the combustion process of methane hydrate sediments, so it is necessary to pay attention to the distribution of dissociated water within the pore channels in the Pore area. Figure 16 shows the concentration distribution of dissociated water in the local area of Pore at different particle sizes. From the figure, the concentration of dissociated water in the local area of Pore gradually decreases with the increase in particle size. By comparing Figures 4 and 16, it can be concluded that with the increase in particle size, the concentration of dissociated water in the Pore area decreases gradually, and the heat acquisition mode of methane hydrate

particles gradually changes from the transport heat transfer process of dissociated water to the heat conduction process of gravel. When the particle size is 1.5 mm, the superposition effect of the two heat transfer methods is the weakest, so the 1.5 mm sample has the smallest average flame height and maximum flame height. The heat transfer effect of gravel is inferior to the transport heat transfer effect of dissociated water, so the combustion ratio per unit time gradually decreases as the particle size increases.

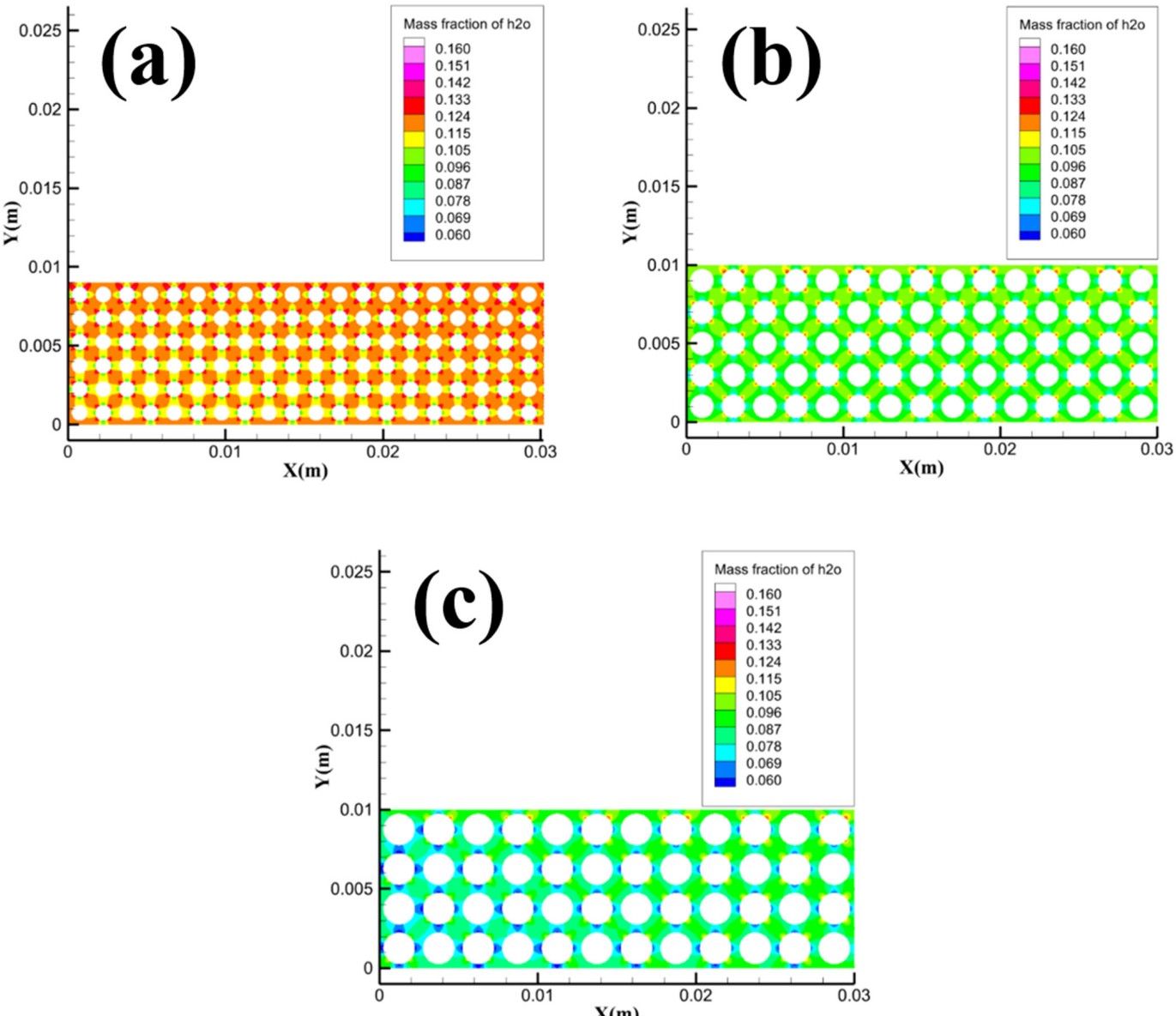

**Figure 16.** Concentration distribution of dissociated water in the local area of Pore under different particle sizes: (**a**) 1 mm, (**b**) 1.5 mm, and (**c**) 2 mm.

### 4.5. Mechanism of Porosity Influence on the Combustion Process of Methane Hydrate Sediments

To investigate the effect of porosity on the combustion process of methane hydrate sediments, three grid models with different porosities, 0.598, 0.667, and 0.805, are constructed. In addition, since the methane inlet velocity has a significant effect on the calculation results of numerical simulations, the methane inlet velocity of the model with different porosities needs to be discussed. Figure 17 shows the variation law of methane released rate with porosity as measured by combustion experiments. From the figure, the methane release rate gradually increases with the increase in porosity. In fact, when the porosity exceeds 0.5,

the difference in methane released rate is about 0.001 g/s for methane hydrate sediment samples formed by quartz sand or brown corundum, that is, the samples have similar methane released rates at high porosity. Therefore, to facilitate the comparison of different combustion conditions and to consider the calculation effect and convergence speed, the methane inlet velocity of the model with different porosities is also set to $2.983 \times 10^{-5}$ m/s.

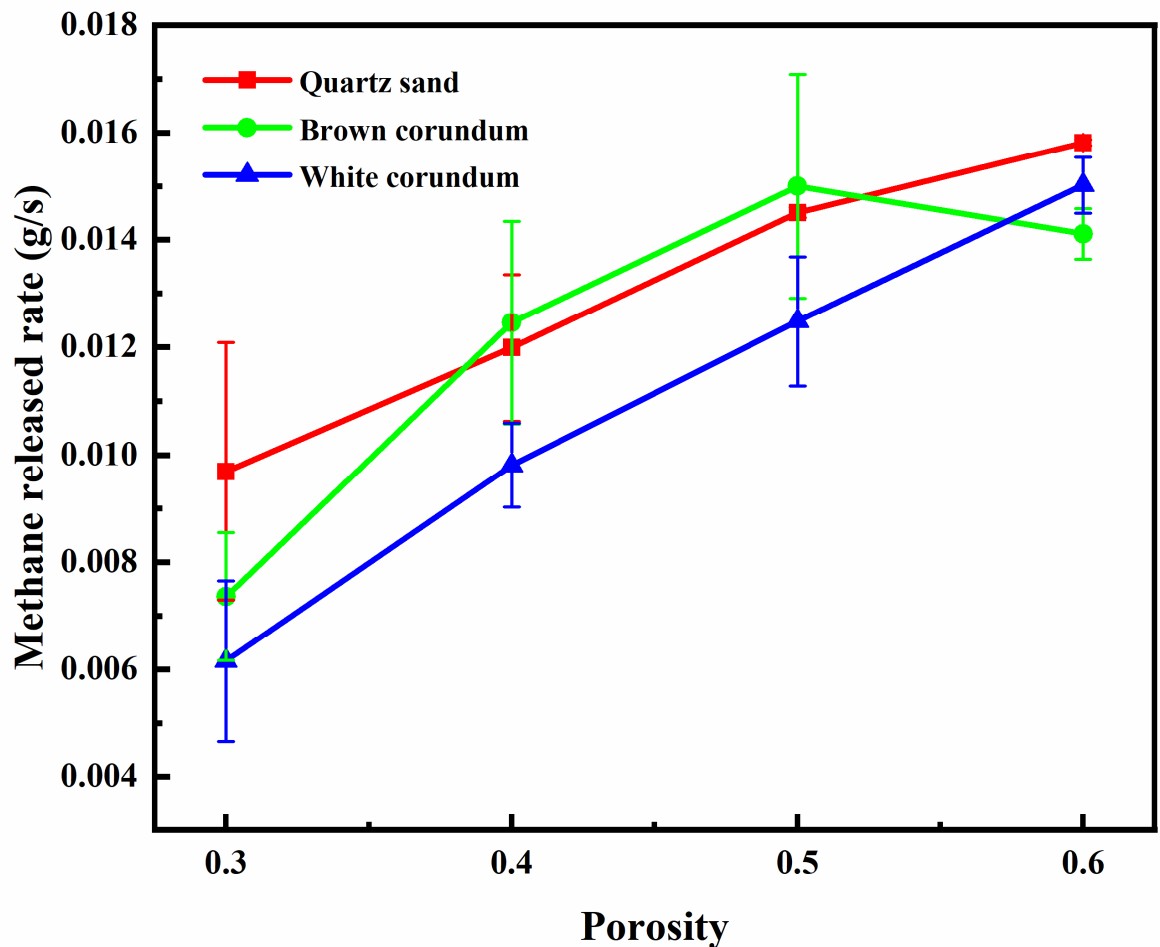

**Figure 17.** Image of methane released rate changing with porosity.

Figure 18 shows the temperature distribution of methane hydrate sediments with different porosities. From the figure, it is observed that with the increase in porosity, the temperature field of the samples does not change significantly, and only the area of the high-temperature reaction zone changes where the temperature exceeds 800 K. The simulation results in Figure 18 match the variation law in Figure 5. When the porosity exceeds 0.5, the average flame height and maximum flame height do not change significantly with the increase in porosity.

Figure 19 shows the temperature distribution in the Pore area at different porosities. From the figure, the temperature variation in the Pore area gradually decreases with the increase in porosity. When the porosity is small, the heat and mass transfer between the sample and the flame is more impeded, and the methane released from the dissociated methane hydrate particles will be temporarily stored in the pore channel. Due to the heat transfer effect of the gravel, the dissociated methane may absorb heat and react with the air entering the pore channel, which in turn leads to a temperature increase in the Pore area. When the porosity is large, the heat and mass transfer between the sample and the flame is less impeded, and the methane released from the dissociated methane hydrate particles does not remain in the pore channels but enters the flame reaction zone directly to participate in the reaction, so there is no significant temperature increase in the Pore area.

In addition, when methane reacts with air in the pore channel, the combustion mode can be considered premixed combustion because the pore channel space is small and the methane can be considered more fully mixed with air. When methane reacts with air in the outer open space (Outer area), the combustion mode can be considered diffusion combustion because the reaction space is large and the methane and air will mix while the chemical reaction occurs. Therefore, by combining Figures 18 and 19, it can be concluded that the combustion mode of methane hydrate sediments gradually shifts from partial premix combustion to diffusion combustion as the porosity increases [39].

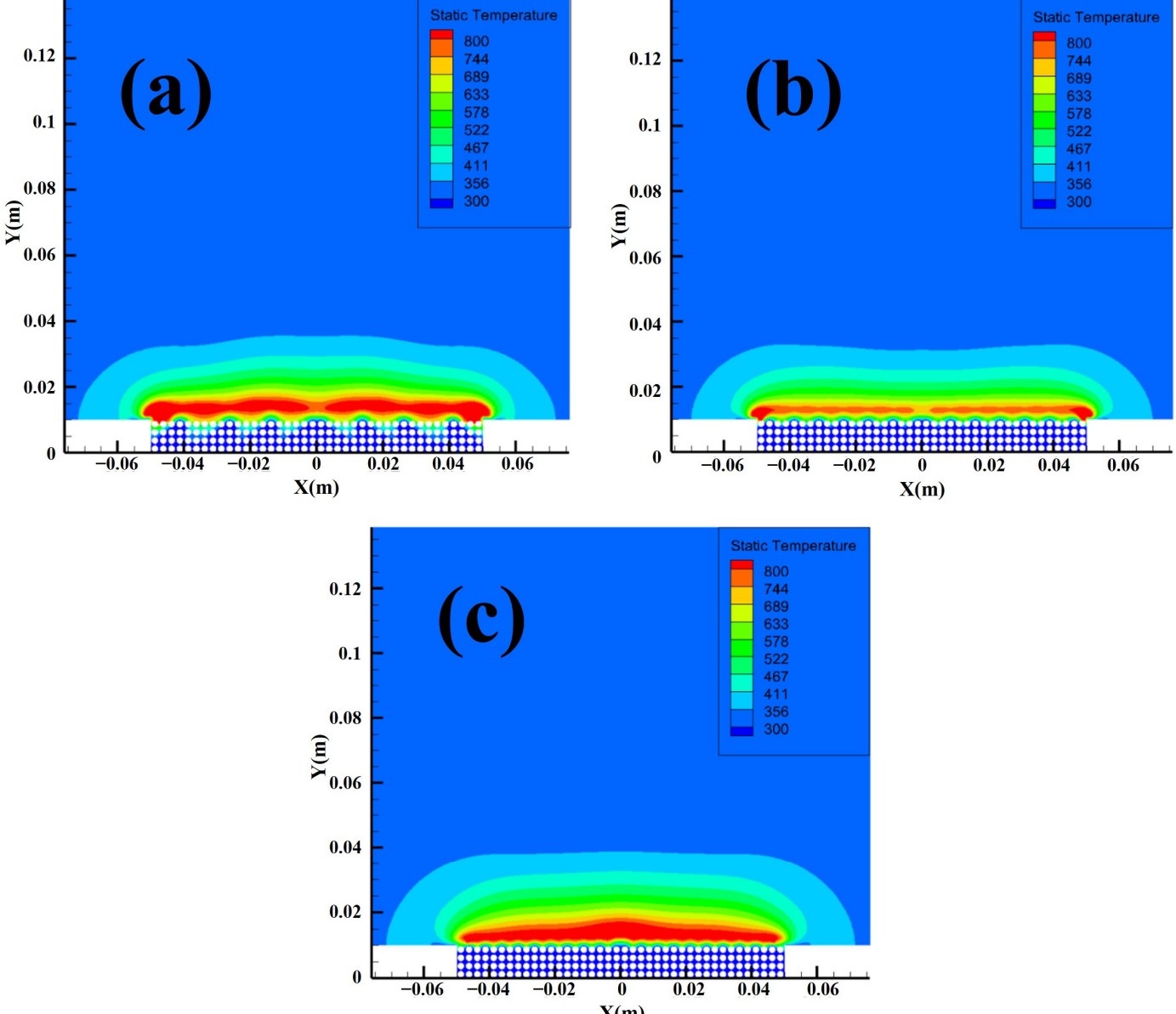

**Figure 18.** Temperature distribution of methane hydrate sediments with different porosities: (**a**) 0.598, (**b**) 0.667, and (**c**) 0.805.

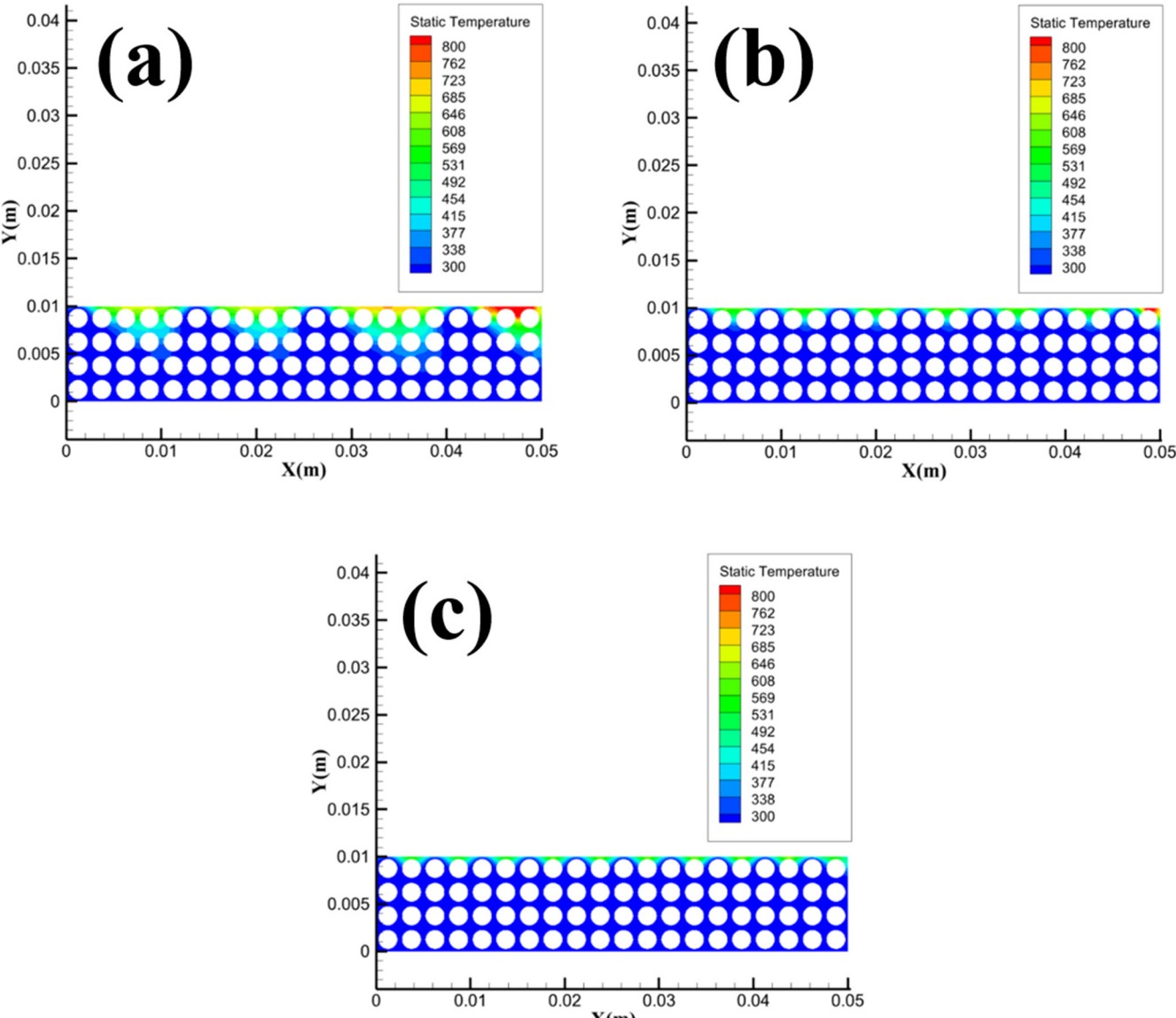

**Figure 19.** Temperature distribution of Pore area under different porosities: (**a**) 0.598, (**b**) 0.667, and (**c**) 0.805.

Figure 20 shows the distribution of dissociated water concentration in the Pore area with different porosities. It can be seen from the figure that the concentration of dissociated water in the pore channel of the sample gradually increases as the porosity increases. The increase in the concentration of dissociated water in the Pore area is not obvious when the porosity increases from 0.598 to 0.667, while the concentration of dissociated water in the Pore area increases significantly when the porosity increases from 0.667 to 0.805. The simulation results in Figure 20 are consistent with the variation law in Figure 7. As the porosity increases, the heat and mass transfer hindrance between the sample and the flame decreases, and the dissociated water in the pore channels can more easily transport and seep to the bottom of the sample, which in turn leads to an increase in the amount of dissociated water absorbed by the combustion tank, and thus the dimensionless discharge water mass increases.

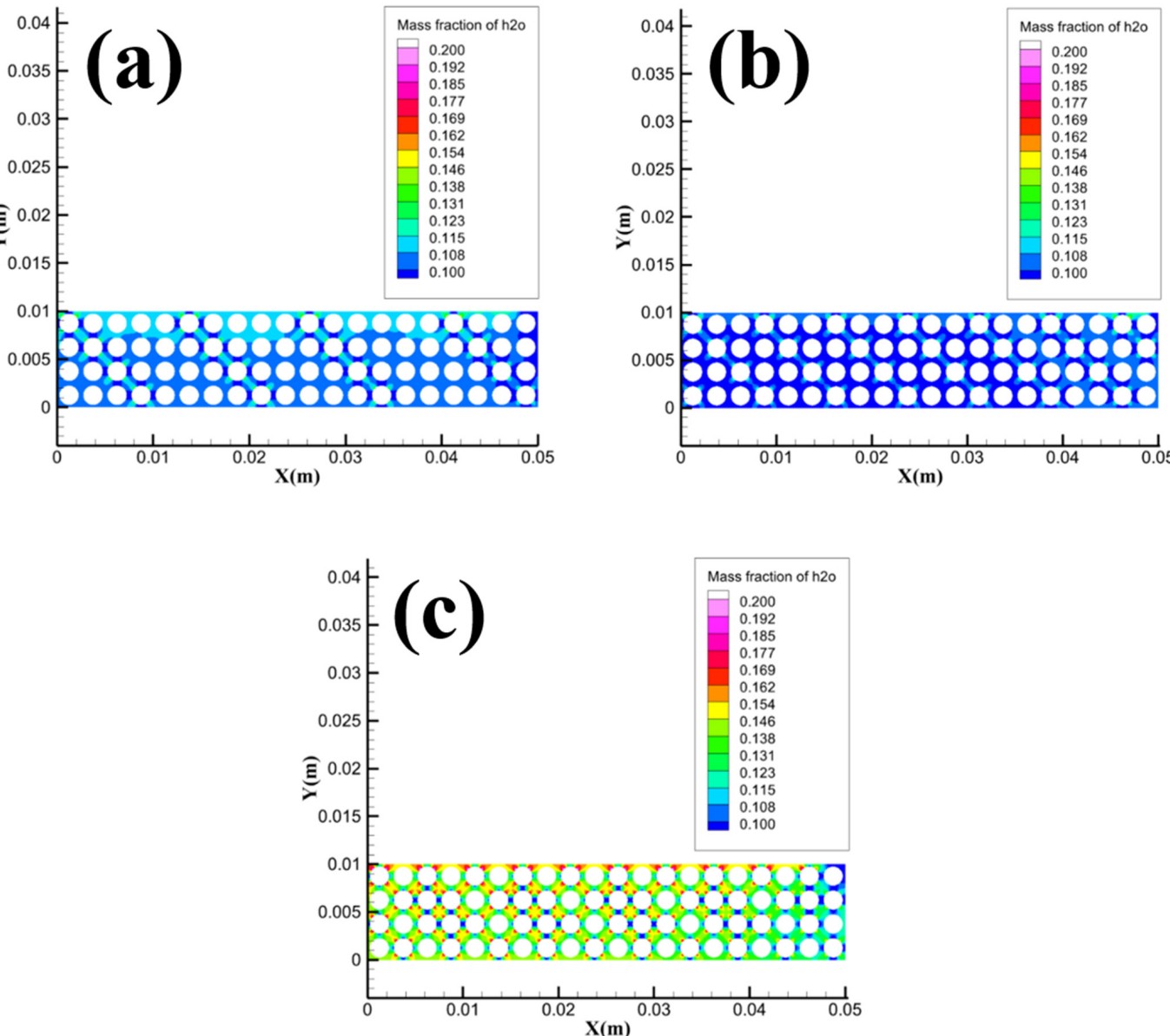

**Figure 20.** Concentration distribution of dissociated water in Pore area under different porosities: (**a**) 0.598, (**b**) 0.667, and (**c**) 0.805.

### 4.6. Mechanism of Saturation Effect on the Combustion Process of Methane Hydrate Sediments

Figure 21 shows the variation law of the methane released rate with saturation, and it can be seen from the figure that the methane released rate increases gradually with the increase in saturation. Therefore, the influence mechanism of different saturations on the combustion process of methane hydrate sediments can be transformed into the influence mechanism of different methane inlet velocities on the combustion process of methane hydrate sediments by adjusting the methane inlet velocity in numerical simulation. Based on the experimental measurement data, three different methane inlet velocities of $1.492 \times 10^{-5}$ m/s, $2.983 \times 10^{-5}$ m/s, and $5.966 \times 10^{-5}$ m/s are set to investigate the mechanism of the effect of different methane release velocities (different saturation) on the combustion of methane hydrate sediments.

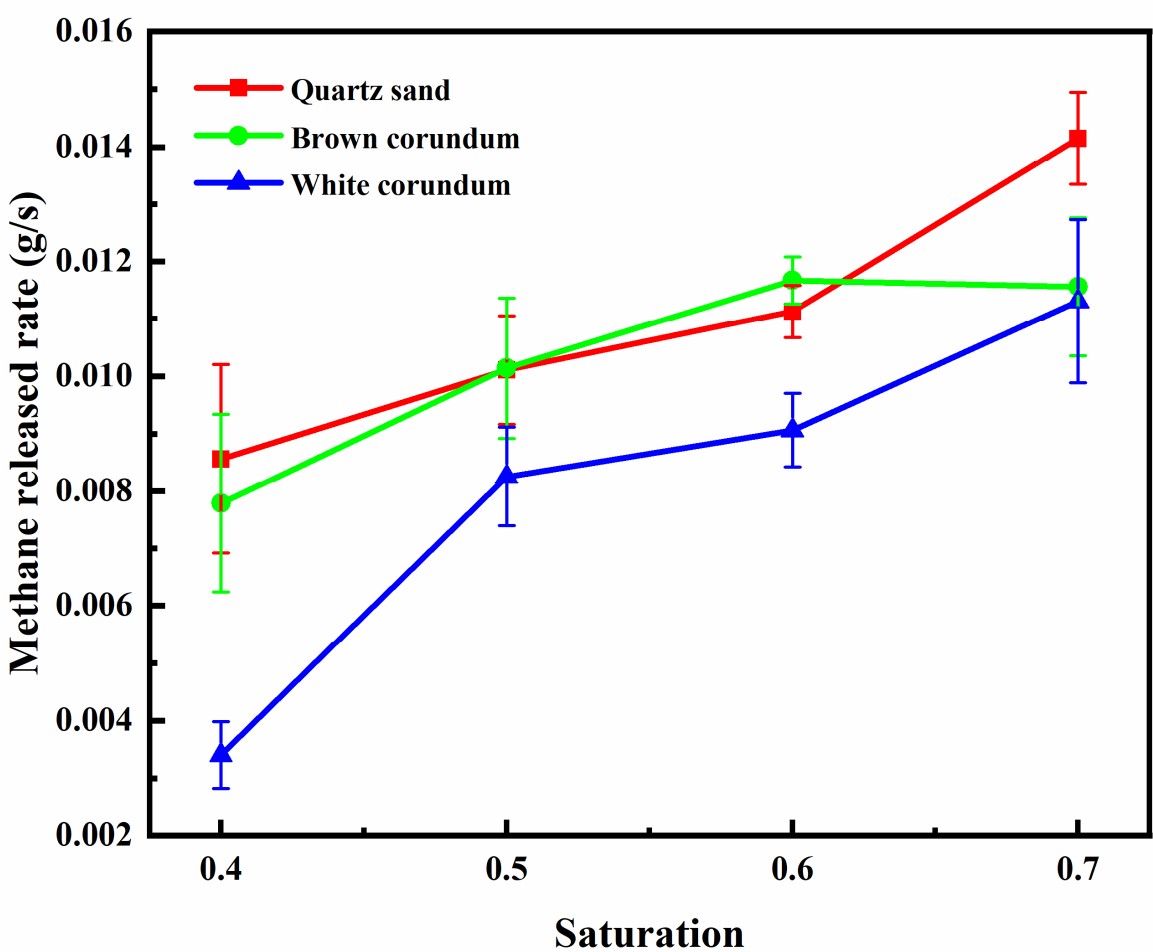

**Figure 21.** Image of methane released rate changing with saturation.

Figure 22 shows the temperature distribution of methane hydrate sediments at different methane inlet velocities. As can be seen from the figure, with the increase in methane inlet velocity, both the amount of methane involved in the reaction during combustion and the momentum of methane increase. Therefore, the area of the high-temperature reaction zone with a temperature over 800 K gradually increases, and the intensity of the combustion reaction increases. When the methane inlet velocity is low, the turbulent energy in the combustion reaction zone is not large, so the temperature field distribution and the high-temperature reaction zone have a regular shape. When the methane inlet velocity is large, the turbulent energy in the combustion reaction zone is large, so the temperature field distribution and the high-temperature reaction zone no longer have a regular shape. Comparing Figure 22 with Figure 8, it can be summarized that as the saturation increases, the rate of methane release from the sample increases, the amount of methane involved in the combustion reaction increases, and the momentum possessed by the methane increases. Therefore, the reaction intensity and turbulent kinetic energy in the high-temperature reaction zone increase, and the flame height is thus enhanced.

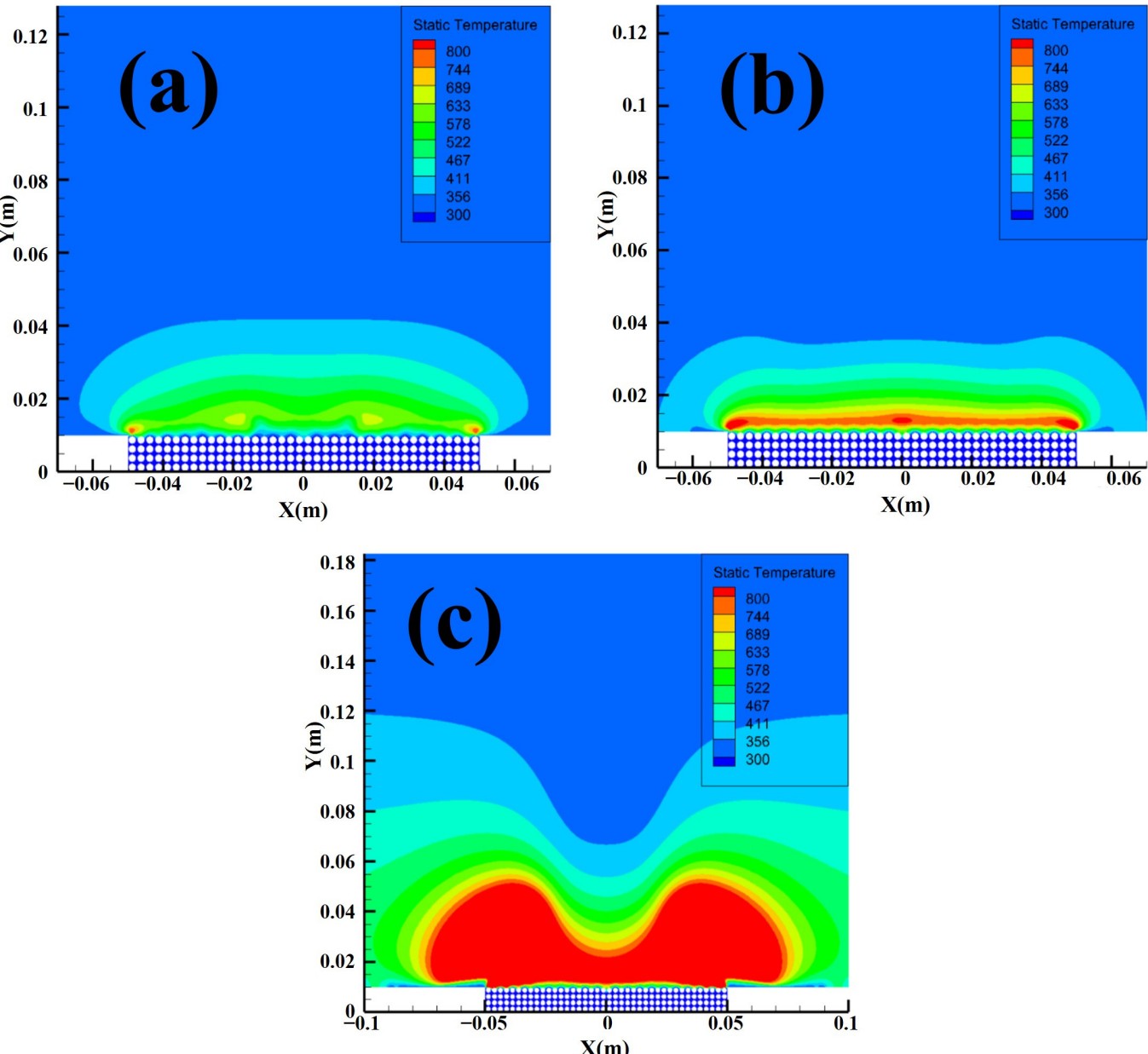

**Figure 22.** Temperature distribution of methane hydrate sediments under different methane inlet velocities: (**a**) $1.492 \times 10^{-5}$ m/s, (**b**) $2.983 \times 10^{-5}$ m/s, and (**c**) $5.966 \times 10^{-5}$ m/s.

Figure 23 shows the methane concentration distribution in the local area of Pore at different methane inlet velocities. From the figure, it can be observed that the methane concentration in the pore channel of the Pore area gradually increases with the increase in methane inlet velocity. Comparing Figure 23 with Figure 9, it can be concluded that as the saturation increases, the concentration of methane in the pore channel increases, and more methane will be involved in the reaction when the flame transfers heat from the sample surface to the middle and bottom of the sample, and the effective combustion ratio and the change in combustion ratio per unit time are thus enhanced.

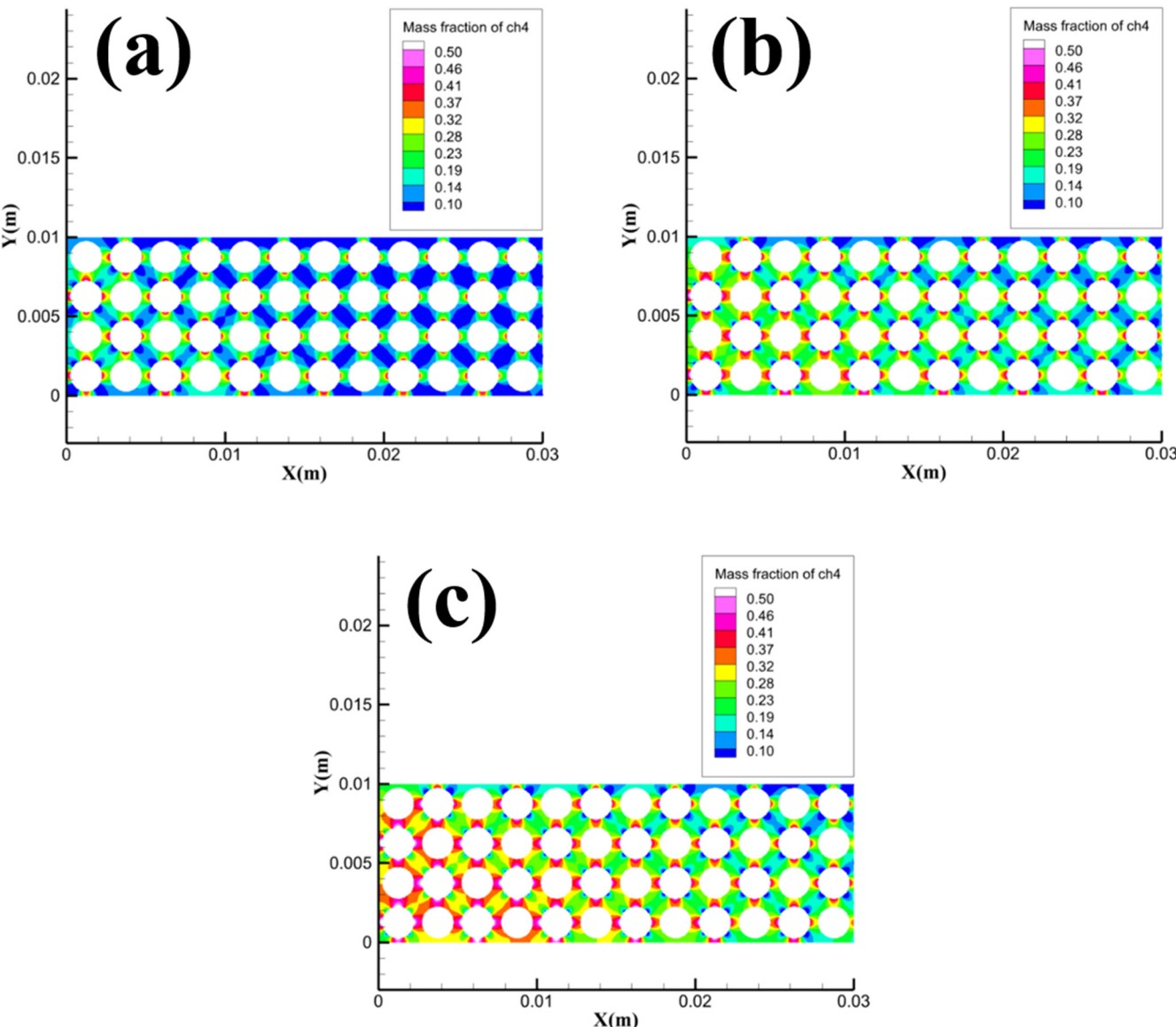

**Figure 23.** Concentration distribution of methane in the local area of Pore under different methane inlet velocities: (**a**) $1.492 \times 10^{-5}$ m/s, (**b**) $2.983 \times 10^{-5}$ m/s, and (**c**) $5.966 \times 10^{-5}$ m/s.

Figure 24 shows the distribution of dissociated water in the local area of Pore at different methane inlet velocities. It should be noted that the values of the mass fraction of dissociated water in the figure range from 0.01 to 0.15 when the inlet velocity of methane is $1.492 \times 10^{-5}$ m/s and $2.983 \times 10^{-5}$ m/s, and when the inlet velocity of methane is $5.966 \times 10^{-5}$ m/s, the values of the dissociated water mass fraction in the figure range from 0.2 to 0.4. Even though the values of the mass fraction are different, it is obvious to compare that the concentration of the dissociated water at the velocity of $5.966 \times 10^{-5}$ m/s is much larger than that at the velocities of $1.492 \times 10^{-5}$ m/s and $2.983 \times 10^{-5}$ m/s. By combining Figures 10 and 24, it can be concluded that when the saturation does not exceed 0.6, the dimensionless discharge water mass first decreases slightly or remains basically unchanged with the increase in saturation. When the saturation exceeds 0.6, the dimensionless discharge water mass increases sharply with the increase in saturation. When the methane inlet velocity increases from $1.492 \times 10^{-5}$ m/s to $2.983 \times 10^{-5}$ m/s, the concentration of dissociated water in the pore channel increases slightly. When the individual methane inlet velocity increases from $2.983 \times 10^{-5}$ m/s to $5.966 \times 10^{-5}$ m/s, the concentration of dissociated water in the pore channel increases significantly.

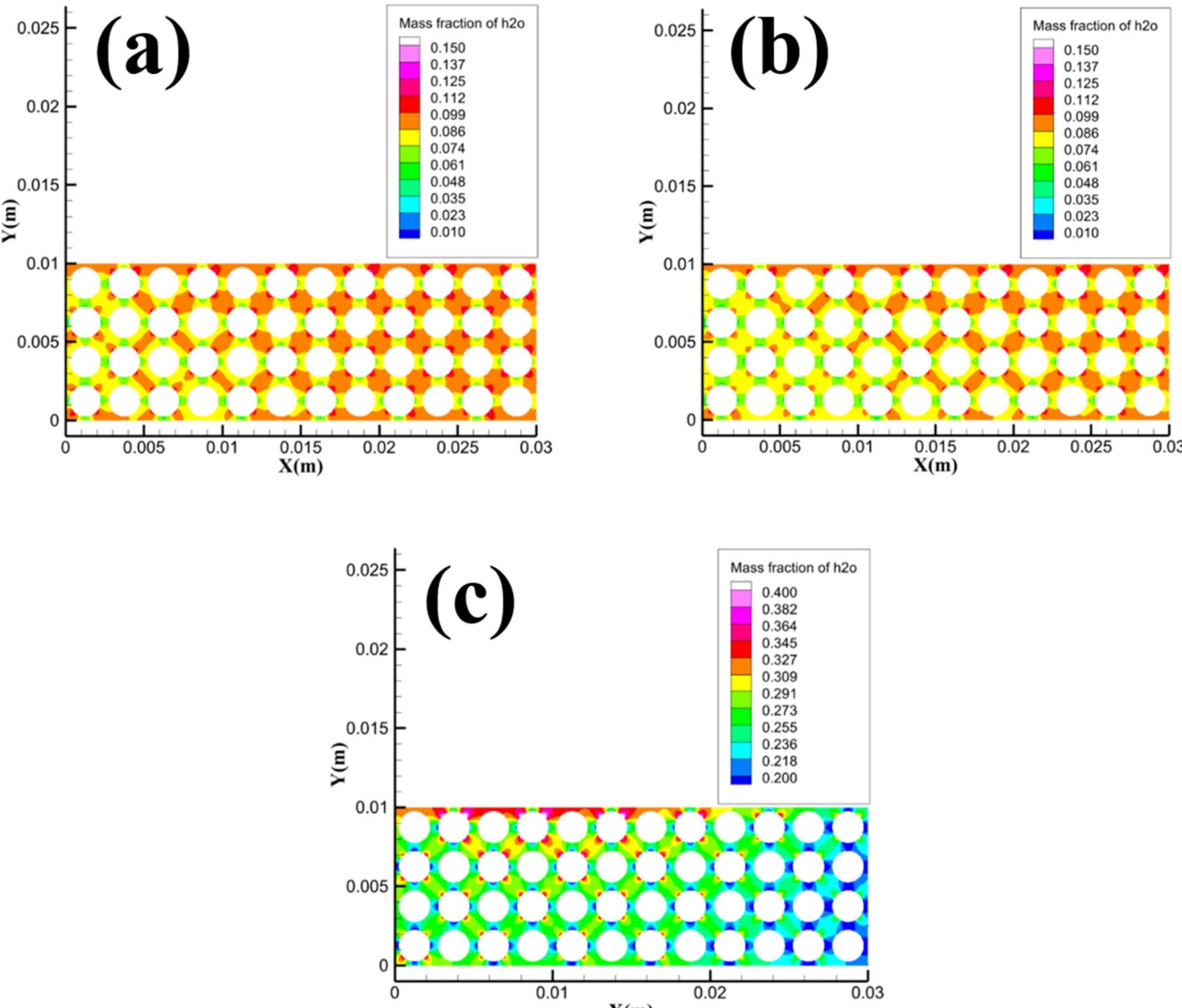

**Figure 24.** Concentration distribution of dissociated water in the local area of the Pore under different methane inlet velocities: (**a**) $1.492 \times 10^{-5}$ m/s, (**b**) $2.983 \times 10^{-5}$ m/s, and (**c**) $5.966 \times 10^{-5}$ m/s.

## 5. Conclusions

(1) The increase in thermal conductivity will lead to an increase in heat dissipation of flame, which will lead to a significant decrease in combustion characteristic parameters.

(2) With the increase in particle size, the way methane hydrate particles obtain heat gradually changes from the migration and heat transfer process of dissociated water to the heat conduction process of gravel, which leads to the minimum value of combustion characteristic parameters at 1.5 mm.

(3) With the increase in porosity, the combustion mode of methane hydrate deposits gradually changes from partial premixed combustion to diffusion combustion, and the combustion characteristic parameters show an increasing trend.

(4) With the increase in saturation, the number of methane molecules participating in the combustion reaction increases, and methane has more momentum. Therefore, the reaction intensity and turbulent kinetic energy in the high-temperature reaction zone increase, and other combustion characteristic parameters except dimensionless discharge water mass show an increasing trend.

(5) To optimize the combustion of methane hydrate sediments, it is recommended to use methane hydrate sediment samples with high saturation and low thermal conductivity, while the oxidant concentration and porosity of methane hydrate sediment samples should be increased.

## 6. Outlook

The actual storage of hydrates is quite complex. Based on the progress in related research areas, future research can focus on the following aspects:

(1) The storage of hydrates often involves high-pressure environments, whereas this study focuses on methane hydrate combustion under atmospheric pressure. Therefore, further research is needed to investigate the combustion characteristics of methane hydrates under high-pressure conditions.

(2) In the in-situ combustion extraction method, the use of methane hydrate sediments from in-situ reservoirs as fuel is still in the experimental simulation and trial extraction stages. The technology is not yet mature, and the challenge of igniting hydrate sediments in situ remains to be resolved.

(3) During the combustion of methane hydrate sediments, there is significant heat absorption due to the accumulation of meltwater, leading to substantial heat loss. Therefore, forced dewatering can be an effective optimization method for combustion, and it is crucial for achieving efficient methane hydrate extraction. Currently, research in this area is relatively limited, and how to remove water from hydrate reservoirs during methane hydrate extraction remains a significant challenge.

(4) The ignition process of hydrate sediments carries certain risks. Ensuring the safety and reliability of hydrate combustion extraction, controlling the combustion of hydrates within manageable limits, and preventing accidents such as explosions remain important issues that need to be addressed.

**Author Contributions:** Conceptualization, G.C.; methodology, Y.L., S.Y., T.G., J.Y. and X.X.; software, D.W.; validation, D.W. and Y.L.; formal analysis, D.W.; investigation, Y.L., S.Y., T.G., J.Y. and X.X.; resources, G.C. and J.L.; data curation, D.W. and Y.L.; writing—original draft preparation, D.W.; writing—review and editing, G.C.; visualization, D.W.; supervision, J.L.; project administration, G.C.; funding acquisition, G.C. All authors have read and agreed to the published version of the manuscript.

**Funding:** This research was funded by Shandong Provincial Natural Science Foundation of China grant number ZR2023ME088, the National Natural Science Foundation of China grant number 51804329 and Shandong Provincial Natural Science Foundation of China grant number ZR2019QEE009.

**Data Availability Statement:** Data available upon request.

**Acknowledgments:** The authors wish to express their appreciation to the reviewers for their valuable suggestions, which significantly enhanced the presentation of this paper.

**Conflicts of Interest:** The authors declare that they have no known competing financial interests or personal relationships that could have appeared to influence the work reported in this paper.

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
