# Peer review of "Effect of Lithological Parameters on Combustion Characteristics of Methane Hydrate Sediments"

_fire, doi:10.3390/fire6120463_

Round 1
Reviewer 1 Report
Comments and Suggestions for Authors
Review on the manuscript
“Effect of Lithological Parameters on Combustion Characteristics of Methane Hydrate Sediments”
The manuscript investigated the four lithological parameters, i.e. thermal conductivity, particle size, porosity and saturation, are investigated by combining experimental observations with numerical simulations to study the influence laws and mechanisms of action on the combustion process of methane hydrate sediments. Combining the experimental results and simulation calculations, to optimize the combustion of methane hydrate sediments, it is recommended to use methane hydrate sediment samples with high saturation and low thermal conductivity, while the oxidant concentration and porosity of methane hydrate sediment samples should be increased.
The article is of scientific value. The results obtained are new. The article may be accepted after minor changes.
1. In the introduction, the literature review can be slightly expanded the following articles: Combustion Characteristics of Methane Hydrate Flames, Energies 12 (2019) 1939. Experimental investigation of flame spreading over pure methane hydrate in a laminar boundary layer, Proc. Combust. Inst. 34 (2013) 2131–2138. Dissociation of various gas hydrates (methane hydrate, double gas hydrates of methane-propane and methane-isopropanol) during combustion: Assessing the combustion efficiency, Energy 206 (2020) 118120. The influence of key parameters on combustion of double gas hydrate, J. Nat. Gas Sci. Eng. 80 (2020) 103396.
2. Please describe the experimental methodology in detail.
3. Was aggregation of methane hydrate particles observed during storage and during dissociation in experiments?
4. Did the porosity of the methane hydrate layer change during the dissociation process?
Reviewer 2 Report
Comments and Suggestions for Authors
The manuscript explores the impact of thermal conductivity, particle size, porosity, and saturation on the combustion of methane hydrate sediments. Modeling of the combustion process of a porous hydrated layer was carried out. The flame height, effective combustion ratio, and dimensionless discharge water mass are studied. To optimize the methane hydrate sediments combustion, it is recommended to use samples with high saturation and low thermal conductivity. The oxidant concentration and porosity of methane hydrate sediment should be increased.
The article is written clearly. The experimental data and simulation are carried out carefully. The results obtained are of interest for optimizing the combustion process of methane hydrate sediments.
As part of a minor revision, some issues need to be clarified before publication of the manuscript.
1. In Figure 4, why does the combustion ratio first decrease until the particle size is 1.5 and then increase with the size 2.0?
2. In Figure 5, why the standard deviation is higher for brown corundum?
3. Part 4.2 - which experiment does it repeat? Please, make a proper reference or give more explanations here.
4. What is your contribution to this model? Did you develop it? what phenomena and effects can be accounted for here? We see the change in the temperature distribution - how do you heat it?
5. Was your model validated?
6. Conclusions: Which part of these conclusions are made based on experimental and what on the numerical model? What was the reason for this work? How your assumptions and experimental and numerical results are matched? Wasn't it the purpose of the work?
Comments on the Quality of English LanguageMinor editing of English language required for this manuscript.
